# Deciphering metabolic differentiation during *Bacillus subtilis* sporulation

Juan D. Tibocha-Bonilla[1,8], Jelani Lyda [2,8], Eammon Riley[2,7], Kit Pogliano [2] ✉ & Karsten Zengler [3,4,5,6] ✉

The bacterium *Bacillus subtilis* undergoes asymmetric cell division during sporulation, producing a mother cell and a smaller forespore connected by the SpoIIQ-SpoIIIA (or Q-A) channel. The two cells differentiate metabolically, and the forespore becomes dependent on the mother cell for essential building blocks. Here, we investigate the metabolic interactions between mother cell and forespore using genome-scale metabolic and expression models as well as experiments. Our results indicate that nucleotides are synthesized in the mother cell and transported in the form of nucleoside di- or tri-phosphates to the forespore via the Q-A channel. However, if the Q-A channel is inactivated later in sporulation, then glycolytic enzymes can form an ATP and NADH shuttle, providing the forespore with energy and reducing power. Our integrated in silico and in vivo approach sheds light into the intricate metabolic interactions underlying cell differentiation in *B. subtilis*, and provides a foundation for future studies of metabolic differentiation.

*Bacillus subtilis* is a model organism for studying bacterial sporulation, a process where a cell forms dormant, resistant endospores[1–4]. Sporulation involves an asymmetric cell division, producing a larger mother cell and smaller forespore, each regulated by tight, cell-specific gene expression. The mother cell engulfs the forespore, which then transitions to dormancy[1]. After engulfment, the forespore grows three-fold in volume[5], develops a protective coat and cortex, and dehydrates. The mother cell eventually lyses to release the mature spore, which can remain dormant for years. When the spore detects a sufficient nutrient concentration, the spore germinates and re-enters the vegetative cycle of growth[2,6,7]. This process requires precise metabolic interactions that are currently not fully understood.

A recent study showed that spore formation involves a dramatic metabolic differentiation of the mother cell and forespore during which enzymes in central metabolism, the TCA cycle, amino acid biosynthesis that are produced prior to asymmetric division are actively depleted from the forespore[8]. This disables forespore metabolism thereby making the forespore dependent on mother cell produced building blocks for the protein synthesis that is required to complete spore assembly. The publication used spatiotemporally-regulated proteolysis (STRP), which specifically depletes tagged proteins from the mother cell, the forespore, or both cells at defined stages of development[9] to demonstrate that dozens of enzymes required to produce metabolic building blocks are essential in the mother cell but not in the forespore. The study also demonstrated that the SpoIIQ-SpoIIIA (Q-A) complex that spans both cells[10] is required for forespore protein synthesis and allows the movement of calcein between the cells, indicating that it likely assembles a passive channel that allows metabolites to move between the mother cell and forespore. These studies provide strong support for the long-standing hypothesis that these proteins assemble a channel that allows the mother cell to nurture the forespore[10–13], and they are consistent with proteomic studies showing that mature spores are deficient in key metabolic enzymes[14,15]. However, the full extent of metabolic

[1]Bioinformatics and Systems Biology Graduate Program, University of California, San Diego, 9500 Gilman Drive, La Jolla, CA, USA. [2]School of Biological Sciences, University of California, San Diego, 9500 Gilman Drive, La Jolla, CA, USA. [3]Department of Pediatrics, University of California, San Diego, 9500 Gilman Drive, La Jolla, CA, USA. [4]Shu Chien - Gene Lay Department of Bioengineering, University of California, San Diego, 9500 Gilman Drive, La Jolla, CA, USA. [5]Center for Microbiome Innovation, University of California, San Diego, 9500 Gilman Drive, La Jolla, CA, USA. [6]Program in Materials Science and Engineering, University of California, San Diego, 9500 Gilman Drive, La Jolla, CA, USA. [7]Present address: Ginkgo Bioworks, Inc., Boston, MA, USA. [8]These authors contributed equally: Juan D. Tibocha-Bonilla, Jelani Lyda. ✉e-mail: kpogliano@ucsd.edu; kzengler@ucsd.edu

differentiation and metabolite exchange remains unclear, since metabolism involves thousands of enzymes and metabolites for which similar assays are not feasible. Furthermore, STRP is challenging or impossible for genes in operons and for proteins whose C-terminus is extracellular or occluded by protein-protein interactions. Thus, we developed a predictive systems biology approach, using metabolic and gene expression models (ME-models) to investigate interactions, testing the cell-specific requirement of different enzymes using GFP-tagging for protein abundance determination, and STRP for targeted degradation.

Our strategy is leveraged by the vast availability of *B. subtilis* genome annotations that led to high-quality genome-scale metabolic (M-) and gene expression (ME-) models. M-models describe metabolism and its responses to perturbations using metabolic fluxes[16], while ME-models also integrate gene expression[17]. These unique ME-model features facilitate the study of processes with tightly regulated gene expression, such as the mother cell and forespore, which have independent genetic programs and unique metabolisms. While M-models have been used to study the metabolic interactions of two or more organisms[18], multi-cell ME-models have remained largely unexplored, with only one study generating a multi-strain *E. coli* community ME-model to design syntrophic co-cultures[19].

Here, we describe comprehensive metabolic interactions between the mother cell and the forespore during spore formation through model-guided STRP and GFP-tagging experiments. We employed a ME-model of *B. subtilis*, *i*JT964-ME[20], to generate a two-cell ME-model (ME2-model), SporeME2, that includes the independent gene expression of the mother cell and the forespore, depletion of proteins from the forespore, and the metabolic interactions between the two cells. Model-generated hypotheses were used to design experiments and identify the metabolic exchanges that provide the forespore with biomass precursors and energy. Finally, we used SporeME2 to contextualize dozens of protein depletions or dilutions in the forespore that have been suggested by mass spectrometry and fluorescence microscopy. The model accurately predicts the impact of depleting specific enzymes from the forespore on the activity of other metabolic pathways in both cells, providing a proteome-wide view of metabolic differentiation in the two cells required to assemble a spore.

## Results

### Properties of the SporeME2 model of *Bacillus subtilis* for elucidating sporulation

We adapted the existing *B. subtilis* ME-model, *i*JT964-ME[20], to build a ME2-model representing the connected mother cell and forespore (Fig. 1a). Each model contained a metabolic network and gene expression network (Fig. 1b), with the forespore model excluding 13 proteins identified to be depleted in the forespore by mass spectrometry and validated through a GFP localization assay to be depleted in the forespore but present in the mother cell[21] (data are available via ProteomeXchange[22] with identifier PXD051727) (Fig. 1c, Supplementary Fig. 1). Furthermore, we allowed for transport of all metabolic intermediates via the sporulation-specific SpoIIQ-SpoIIIA complex (Q-A)[10–13,23,24] that has been shown to facilitate transport of calcein[8], which is larger than most metabolic intermediates. In brief, the mother cell

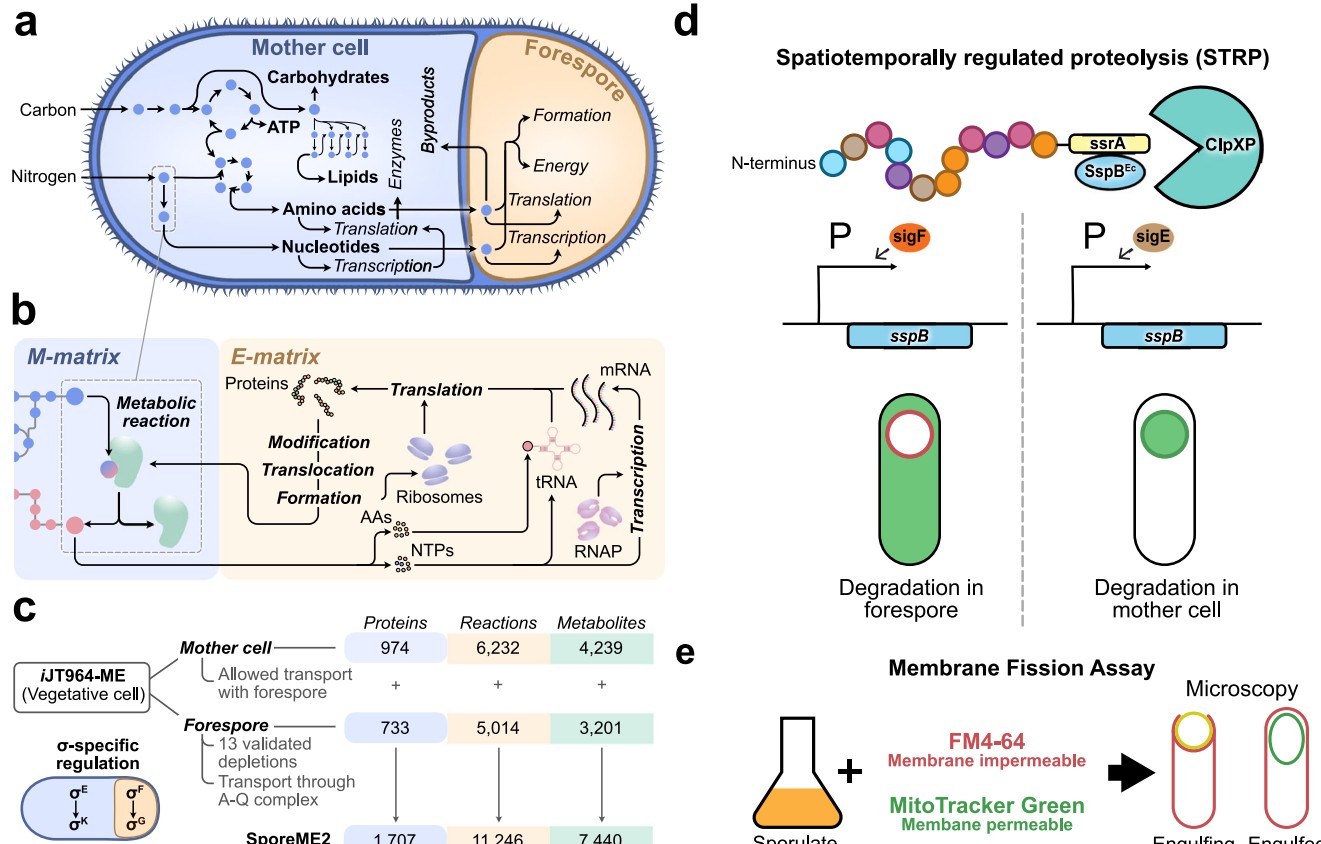

**Fig. 1 | Integrated modeling and experimental approach for the understanding of metabolic exchanges between mother cell and forespore. a** Metabolic nurturing model of the mother cell and the forespore during spore formation as described by Riley et al.[8]. **b** Integration of gene expression with metabolism in ME-models, and hence in SporeME2. **c** Reconstruction and properties of SporeME2. **d** Spatiotemporally regulated proteolysis (STRP) method. **e** Diagram of membrane fission assay to monitor engulfment by fluorescence microscopy. FM 4-64 (red) is membrane impermeable and stains the forespore membrane during (left) but not after (right) engulfment. Thus, after membrane fission, the forespore membranes are only stained by the membrane permeable stain MitoTracker Green, therefore appearing green by fluorescence microscopy.

and the forespore ME-models inherited the stoichiometric matrix from *i*JT964-ME. Furthermore, expression reactions for 13 identified protein depletions were closed in the forespore ME-model, which was then connected with the mother cell by transport reactions through the Q-A complex.

In total, SporeME2 encompasses 1707 proteins, 11,246 reactions, and 7440 metabolites (Fig. 1c). A complete summary of metabolite and reaction categories and the model's reactions with flux predictions is provided in the Supplement (Supplementary Fig. 1 and Supplementary Data 1). SporeME2 predicts metabolite exchanges by weighing the cost of synthesis in the mother cell and transport to the forespore against the cost of synthesis (if feasible) in the forespore, given that metabolic differentiation disables numerous core pathways in the forespore. In this model, forespore growth rate is used as a proxy for the complete process of forespore formation or spore assembly. However, it is worth noting that, as a ME-model, SporeME2 represents forespore formation through the biosynthesis of all individual biomass components and does not account for the physical process of structural assembly.

## STRP and GFP-tagging interrogate mother cell and forespore metabolism independently

The model's flux predictions were then tested by in vivo analyzes. Protein localization in the mother cell and forespore was confirmed through C-terminal GFP-tagging of enzymes and subsequent fluorescence microscopy. Protein essentiality in one or both cells was assessed through STRP-mediated depletion of enzymes in the mother cell or forespore, allowing us to separate vegetative phenotypes from sporulation phenotypes and test the requirement in each cell individually[9]. This method depends on the ClpXP protease that is present in both cells[25], and the observation that the *E. coli* SspB adapter can direct proteins tagged with the *E. coli* recognition sequence (ssrA*) to *B. subtilis* ClpXP for degradation. Expressing the SspB adapter from forespore- and mother cell-specific promoters thereby allows proteins to be specifically depleted in a cell and stage specific manner[9], enabling identification of proteins required specifically in the mother cell and/or forespore (Fig. 1d). Furthermore, we assessed the ability of cells to complete early stages of sporulation using a membrane fission assay[26] (Fig. 1e) and monitored completion of sporulation by phase bright spores formation, spore viability, and germination.

First, we evaluated how nucleotides are provided to the forespore. There are no annotated transporters for nucleotides in their di- and triphosphate forms (NDP and NTP)[20,27], but it is possible that Q-A allows nucleotide exchange in various phosphorylation states. Thus, we designed simulations selectively allowing and blocking the transport of NDPs or NTPs through Q-A to predict spore defects in each case. In the case of no NDP or NTP transport, nucleotides are predicted by SporeME2 to be synthesized in their monophosphate form (NMP) in the mother cell and then transported to and phosphorylated in the forespore. This is supported by Gmk-GFP and Cmk-GFP (NDP synthesis) being more abundant in the mother cell (Supplementary Data 1) but present in the mother cell and the forespore (Supplementary Fig. 2), which would allow phosphorylation of NMPs to occur in both cells. Transcription of *gmk*, *cmk* and *pyrG* has also been previously reported before and after sporulation[28]. However, if NDPs and NTPs are transported across the septum, then the kinases required to produce NTPs would be dispensable in the forespore for spore assembly, although they would likely be required for spore germination and outgrowth because they are essential proteins (NDP or NTP transport case in Fig. 2c). In contrast, if only NMPs are transported, then the kinases would be required in the forespore for spore assembly (no transport case in Fig. 2c).

We used STRP to test this hypothesis, incorporating SporeME2 predictions. Specifically, we performed STRP on Cmk (phosphorylation of CMP and UMP to CDP and UDP, respectively), PyrG (synthesis of CTP from UTP), and Gmk (phosphorylation of GMP to GDP). The NDPs

are then phosphorylated to CTP, UTP, and GTP by Ndk (Fig. 2a). In addition, SporeME2 predicted that while degrading Gmk will eliminate GMP and GDP phosphorylation, degrading both Cmk and PyrG simultaneously will be required to block CMP, CDP, UMP, and UDP phosphorylation.

To test the model's predictions, we created strains carrying various ssrA-tagged proteins in PY79 *B. subtilis* cells: Gmk-ssrA, for GDP synthesis, and Cmk-ssrA, PyrG-ssrA, and the Cmk-ssrA + PyrG-ssrA double depletion strain for CDP and UDP synthesis. The proteins were degraded early in sporulation in the mother cell only, in the forespore only, or in both cells, and the sporangia were visualized by fluorescence and phase-contrast microscopy at 3 and 5 hours after sporulation initiation. The membrane fission assay was used to monitor engulfment completion. As predicted by the model, no sporulation defect was observed when Cmk or PyrG were depleted independently in the forespore and mother cell (Supplementary Fig. 3), likely because either enzyme can allow CTP synthesis (Fig. 2a). When Gmk or both Cmk and PyrG were degraded, production of phase bright spores was delayed when the enzymes were degraded in the mother cell, suggesting that spore development was inhibited; while degradation in the forespore produced phase bright spores, suggesting that spore development was not affected (Fig. 2d). Spore titer assays showed that degradation in the mother cell resulted in a 1000-fold reduction in viable spores, demonstrating that mother cell enzymes are required for sporulation. In contrast, protein degradation in the forespore did not affect the production of phase-bright spores. However, the mature spores were unable to produce vegetative cells (Fig. 2d, e). Germination time-lapses of the forespore degradation mutants revealed that the spores germinated but failed to outgrow (Fig. 2f).

The fact that Gmk or Cmk and PyrG degradation in the forespore does not impede spore development (Fig. 2e) suggests that kinase degradation in the forespore is compensated during spore development by the mother cell, which presumably transports either NDPs or NTPs to the forespore to fulfill this requirement. Strains in which these proteins are degraded in the forespore still produced phase-bright spores that initiate germination but did not outgrow, indicating that, as expected, these kinases are essential for vegetative growth. Together, these results suggest that the Q-A channel, or an unidentified channel, transports NMPs, NDPs, or NTPs to the forespore during spore formation. Together, integrated SporeME2/STRP experiments revealed a new class of molecules transported to the forespore to foster its development.

## Glycolytic enzymes transduce ATP to the forespore

It is currently unclear how the forespore acquires energy for biosynthesis while it transitions to dormancy[8]. SporeME2 predicts that if direct ATP transport through Q-A or synthesis through F1FO ATP synthase is allowed, it would cover 99.6% of ATP supply to the forespore (Supplementary Fig. 4). However, prior results have demonstrated that SpoIIQ is degraded after engulfment[29,30], suggesting that the Q-A channel may only be active during engulfment and that the F1FO ATPase is absent from the forespore[31] and from mature spores[14]. We therefore simulated alternative pathways that could supply ATP to the forespore in case ATP transport and F1FO ATP synthase were not available at any stage of spore formation.

In these circumstances, SporeME2 predicts that ATP can be provided to the forespore by glycolytic enzymes running glycolysis in opposite directions in the mother cell and the forespore, with the transport of sugars and pyruvate across the forespore membrane (Fig. 3a). In the mother cell, the model suggests that a high-energy carbohydrate such as fructose-6-phosphate (F6P) is produced through the gluconeogenic pathway, involving PycA, PckA, Eno, Pgm, Pgk, GapB, TpiA, FbaA, and Fbp and that F6P is subsequently transported to the forespore. Other sugars such as fructose-1-phosphate, glucose-6-phosphate, or glucose-1-phosphate can also be produced and

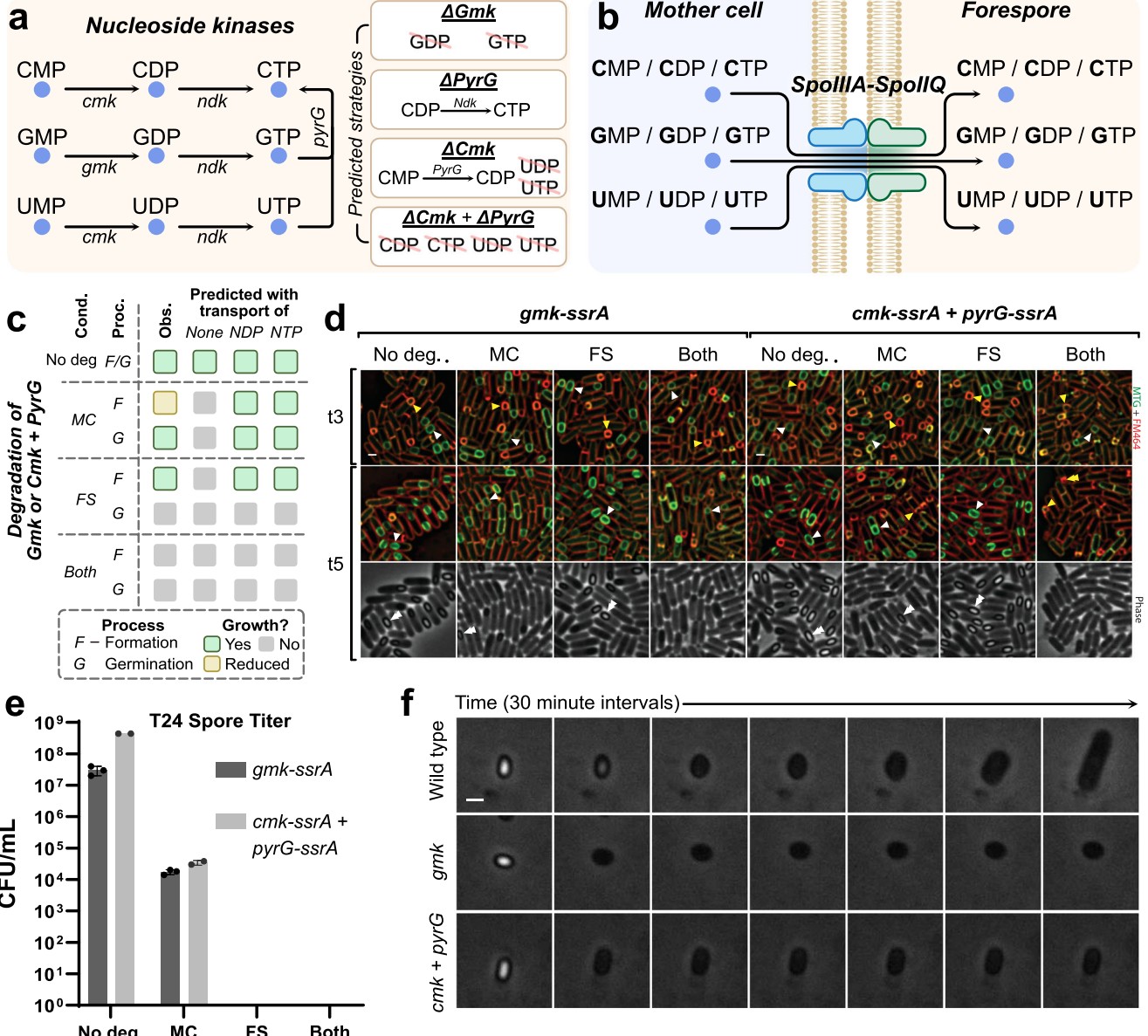

**Fig. 2 | Observations of effects of model-predicted degradation strategies through the STRP system. a** NDP and NTP synthesis through kinases in *B. subtilis* as predicted by SporeME2. **b** NMP, NDP, and NTP transport through the SpoIIIA-SpoIIQ pore channel. **c** Summary of model predicted (with and without allowed NDP/NTP transport) and observed results. A green box represents successful spore formation or outgrowth during germination, while a gray box represents no spore formation or no outgrowth, respectively. **d** Membrane fission assay of *gmk-ssrA* single mutant and *cmk-ssrA + pyrG-ssrA* double mutant at 3 and 5 hours after sporulation initiation using fluorescent and phase microscopy. The tagged proteins were degraded using the STRP system in either the mother cell only [BJAL130, BJAL146] the forespore only [BJAL047, BJAL80], or in both compartments [BJAL132, BJAL152] and compared to the *-ssrA* tagged only strain [KP1638, BJAL070]. The membrane is in red and green. White single arrows indicate mid-engulfment

forespores, and single white arrows indicate completed engulfment forespores. Double yellow arrows indicate lysing membranes. Phase bright endospores are indicated by double white arrows. Scale bar = 1 ɥm. **e** Spore titer of the ssrA-tagged strains 24 hours after sporulation initiation without depletion or after depletion in the mother cell, forespore, or both cells (Supplementary Table 1). Spore titers were measured in three technical replicates for Gmk-ssrA, and two technical replicates for Cmk-ssrA and pyrG-ssrA. The cultures were heated to 80 °C for 20 minutes before being plated. Data are presented as mean values +/- the standard deviation. **f** 3-hour germination timelapse of purified spores from the *-ssrA* tagged mutants using phase microscopy, showing that the spores become hydrated but fail to resume growth. Each time point was taken at 30-minute intervals. Scale bar = 1 ɥm. Source data are provided in the Source Data file.

transported to the forespore (involving Pgi, GlmM, and a transporter), but F6P was predicted as optimal branching point for transport to the forespore. In the forespore, glycolytic enzymes break down F6P to produce NADH and ATP, involving PfkA, FbaA, TpiA, GapA, Pgk, Pgm, Eno, and Pyk. Our simulations predict that, in the absence of ATP transport, this mechanism would account for 72% of the ATP produced in the forespore, mainly through Pgk (50%) and Pyk (22%). The remaining 28% was predicted to be produced by transport from the

mother cell and oxidation of alpha-ketoglutarate (AKG) via the PdhD-OdhAB and the SucCD complexes (Fig. 3a).

The metabolic network of *B. subtilis* would allow for this mechanism to work by transporting phosphoenolpyruvate (PEP) and pyruvate, thus bypassing the upper steps of the gluconeogenic/gly-colytic pathway and LctE in both cells. However, our simulations show that the mechanisms shown in Fig. 3a are beneficial due to the addi-tional production of ATP through Pgk and SucCD. These additional

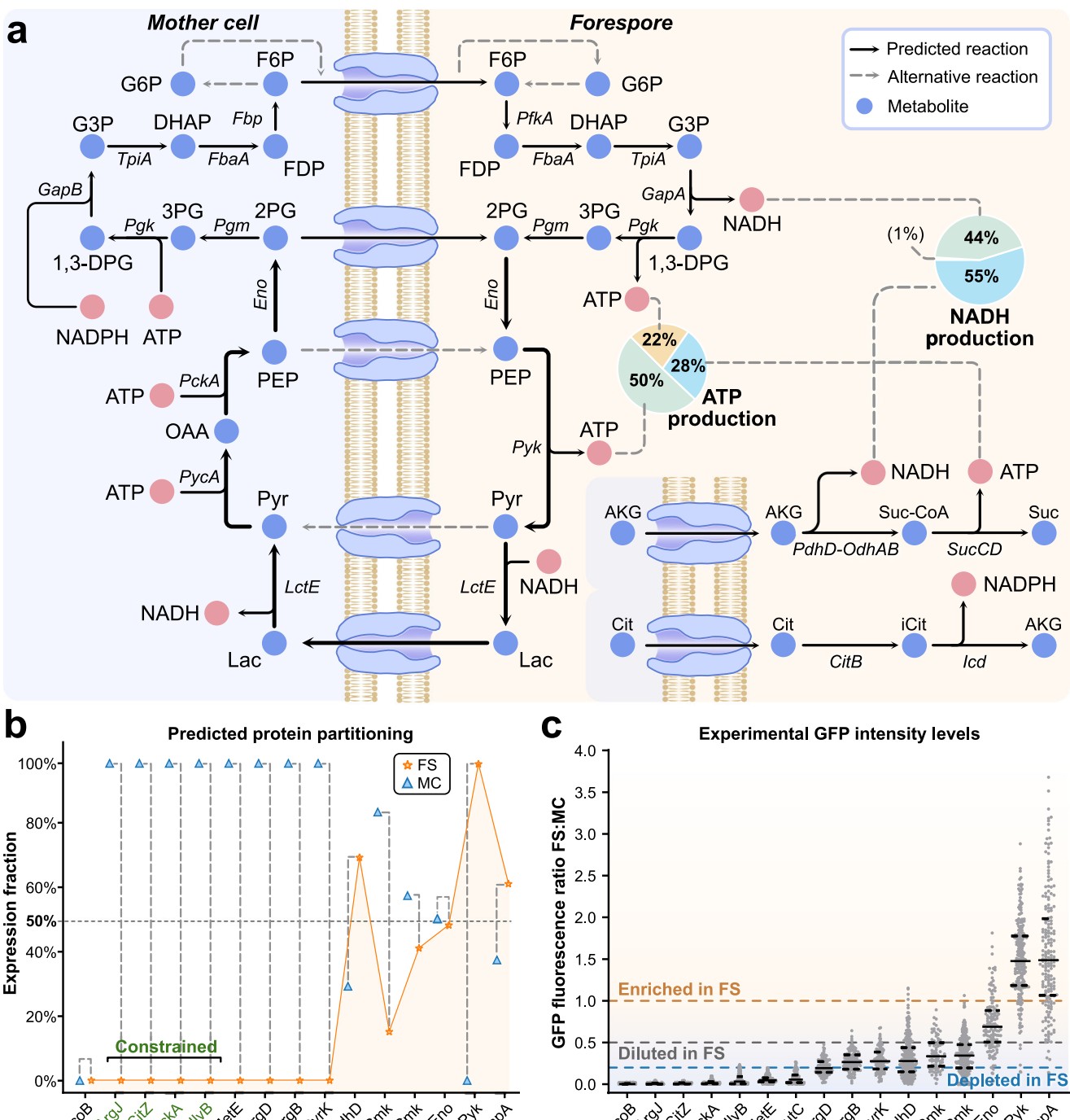

**Fig. 3 | Description of metabolic functions as predicted by *SporeME2* and observed through GFP-tagging. a** Predicted mechanism during forespore formation for the mother cell (MC) to provide the forespore (FS) with energy in the form of ATP and NADH. Glycolytic enzymes were predicted to operate in reverse in the mother cell and forward in the forespore to form an ATP and NADPH/NADH shuttle, producing energy in the forespore with Pgk and Pyk. **b** Predictions by SporeME2 of protein partitioning of core metabolic proteins across the mother cell and the forespore. **c** The forespore-to-mother-cell fluorescence ratio was measured from GFP-tagged proteins in sporulating cells three hours after sporulation initiation. The solid line is the median, while dotted lines are quartiles. The red dotted line = depleted in FS, the blue dotted line = diluted in FS (0.2–0.5), and the purple dotted line = enriched in FS. Measurements were performed in duplicate, in two sessions on the microscope, and four to six images of each strain were analyzed for cell fluorescence. Outliers were removed using Graphpad Prism, ROUT (Q = 0.1%). Source data are provided in the Source Data file.

steps produce an excess of NADH in the forespore by GapA (44%) and PdhD (55%), which is balanced by an additional step of converting pyruvate to lactate via LctE. Moreover, NADPH in the forespore is predicted to be produced through CitB and Icd from mother cell-provided citrate (Fig. 3a). Protein essentiality results from SporeME2 (Supplementary Data 1) highlight that blocking any single mechanism

producing ATP, NADH, or NADPH in the forespore by depleting any protein involved (Fig. 3a) does not prevent forespore formation completely, because other proteins compensate for the lack of the depleted ones (Supplementary Fig. 5) (see essentiality predictions in Supplementary Data 1). Notably, it was previously reported that CitB and Icd are not essential for producing phase-bright spores. The model

further predicted that Pyk and GapA are enriched in the forespore with ratios of 100:0 and 62:38 forespore:mother cell, respectively. Moreover, Eno was required to be distributed with a ratio of 49:51 between the forespore and the mother cell (Fig. 3b) for the ATP shuttle to function.

To test these predictions, we performed GFP-tagging and STRP experiments. The GFP-tagging results (Fig. 3c) support SporeME's predicted ATP shuttle (Fig. 3a), as Pyk and GapA are observed to be significantly enriched in the forespore, while Eno was distributed across both cells. While PdhD was predicted to be enriched in the forespore (ratio 70:30), simulations show its activity in both cells as a subunit of alpha-ketoglutarate dehydrogenase (PdhD-OdhAB) in the forespore and pyruvate dehydrogenase in the mother cell (PdhABCD). GFP-tagging revealed that PdhD was diluted in the forespore, which suggests a lower activity of PdhD-OdhAB in the forespore compared to PdhABCD in the mother cell (Fig. 3c). Therefore, our results suggest that if the Q-A channel is inactivated after engulfment, then energy could be produced in the forespore via a mechanism using glycolytic enzymes.

### Amino acid supply to the forespore is driven by energetics

SporeME2 predicts that the mother cell performs amino acid biosynthesis and feeds them to the forespore, although in a few cases, the transported metabolite is a direct precursor of the final amino acid (Supplementary Data 2). More specifically, the model predicts that alanine, phenylalanine, aspartate, valine, isoleucine, glutamate, glutamine, and cysteine are synthesized in the forespore from precursors such as alpha-ketoglutarate, cystathionine, and monocarboxylic acid, provided by the mother cell.

Notably, certain transported amino acids such as arginine, histidine, tryptophan, tyrosine, and leucine, are among those with the highest Gibbs free energy of formation and biosynthetic cost[32], indicating that these provide the largest nutritional value. Arginine and histidine have been observed in metabolic exchange in co-cultures and communities[33,34], and arginine has been previously shown to be transported from the mother cell to the forespore during spore formation[8]. Other amino acids such as proline, aspartate, asparagine, and threonine, are relatively inexpensive to synthesize[32], but it may be advantageous to perform synthesis in the mother cell rather than in the forespore due to the latter's metabolic slowdown as spore development progresses. GFP intensity levels support this hypothesis, as core biosynthetic enzymes in amino acid metabolism, such as MetE, ArgD, and ArgB, were depleted from the forespore (Fig. 3c), and our MS suggests that 26 amino acid biosynthetic proteins are depleted in the forespore consistent with prior studies[8,14,15]. These results are largely consistent with our simulations, as 25 out of these 26 were predicted inactive in the forespore, with the only exception of IlvC, which serves a role in the interconversion of threonine coming from the mother cell to isoleucine (see translation flux predictions in Supplementary Data 2).

### Protein essentiality predictions reveal metabolic reprogramming at the proteome scale

We leveraged SporeME2 to describe the proteome-scale metabolic reprogramming of the mother cell and the forespore and how this reprogramming compares to the exponential phase vegetative cell. Thus, we used the model to simulate single protein depletions in the forespore, mother cell, and vegetative cell (see Online "Methods") and assessed which depletions prevented spore formation or vegetative growth (i.e., essential proteins) (Fig. 4a). We show the distribution of essential proteins across the forespore, mother cell, and the vegetative cell in Fig. 4a, and the pathway enrichment analysis of these gene groups in Fig. 4b, with raw data available in Supplementary Data 3 and 4, respectively.

There are 176 proteins predicted to be essential in the forespore, 222 in the mother cell, and 322 in the vegetative cell (Supplementary Data 3). The eighty-one proteins predicted to be essential in the mother

cell and vegetative cells but not the forespore were required for biomass precursor and energy production (Fig. 4a, b). In contrast, the 42 proteins predicted to be essential in the forespore and vegetative cells were required mostly for the synthesis of structural components (Fig. 4a) such as the cell wall and membrane (Fig. 4b). The mother cell and the forespore shared a pool of 132 essential proteins with the vegetative cell that were primarily involved in gene expression (Fig. 4a, b), as they perform independent transcription and translation[8].

Sixty-seven required proteins are unique to the vegetative cell, likely involving metabolic pathways that during sporulation can be compensated by either cell if depleted (Fig. 4a, b). The vast majority of proteins (255 out of 266) predicted to be essential in either the forespore or the mother cell were also essential for metabolism in growing cells, with just 11 exceptions (Fig. 4a). Two proteins, the forespore specific transcription factor σ[F] and SpoIIQ, were the only two proteins predicted to be essential only in the forespore, as expected since σ[F] governs the expression of SpoIIQ[35,36], which in turn is an essential component of the Q-A complex[10–13,23,24]. On the other hand, nine proteins were predicted to be essential only in the mother cell, including the eight subunits of SpoIIIAA-AH and the mother cell transcription factor σ[E] that governs their expression. We note that the late transcription factors σ[G] and σ[K] were not found to be essential in this model, likely because they control expression of proteins required for late stages of spore maturation, while the model simulates metabolism and protein synthesis during early forespore formation.

### Predicting new possible depletions at the proteome scale

Next, we investigated the metabolic impact of further depletions at the proteome scale. Seventy-six proteins could be depleted in the forespore, as suggested by MS (PXD051727[21]), but have not been confirmed through GFP-tagging and were not constrained in SporeME2. We therefore calculated whether these depletions would induce the inactivation or blockage of flux through other proteins in specific pathways, with the goal of identifying "keystone" enzymes whose depletion would have the largest impact in disabling spore metabolism. In brief, a protein depletion can induce inactivation of another protein if the latter could perform alternative functions, but the model predicts no translation flux. Furthermore, a blockage occurs when the depletion causes all possible functions of another protein to be infeasible, such as a depletion blocking a linear metabolic pathway (see Online "Methods" for more detailed definitions).

We first identified inactivation and blockage as a result of depletions suggested by MS (PXD051727[21]) (Supplementary Data 5). Our simulations suggest that only three of these depletions will inactivate or block other pathways, namely Mdh, IlvC, and PdhD (Fig. 4d). Numerous proteins are predicted to be inactivated and blocked by them. Thus, if Mdh, IlvC, and PdhD are in fact depleted in the forespore, our predictions suggest that branched-chain amino acid biosynthesis and the steps of glycolysis downstream of Pyk are inactivated.

We repeated the same computational procedure to identify the effect of 13 confirmed depletions (Supplementary Data 6), which required the generation of a naive SporeME2 model with no constrained depletions (see Online "Methods"). Interestingly, only PckA and CitZ were involved in inactivation (either causing or being affected by any), while no direct blockages by them were detected. These results suggest that, out of the 13 confirmed depletions, no single depletion is impactful enough to inactivate or block entire pathways that fully differentiate the metabolism of the forespore. Rather, it is the combination of several depletions that metabolically differentiates the forepore.

## Discussion

Prior studies have used metabolic models for data contextualization and experimental design[16,18], but cell differentiation remained out of

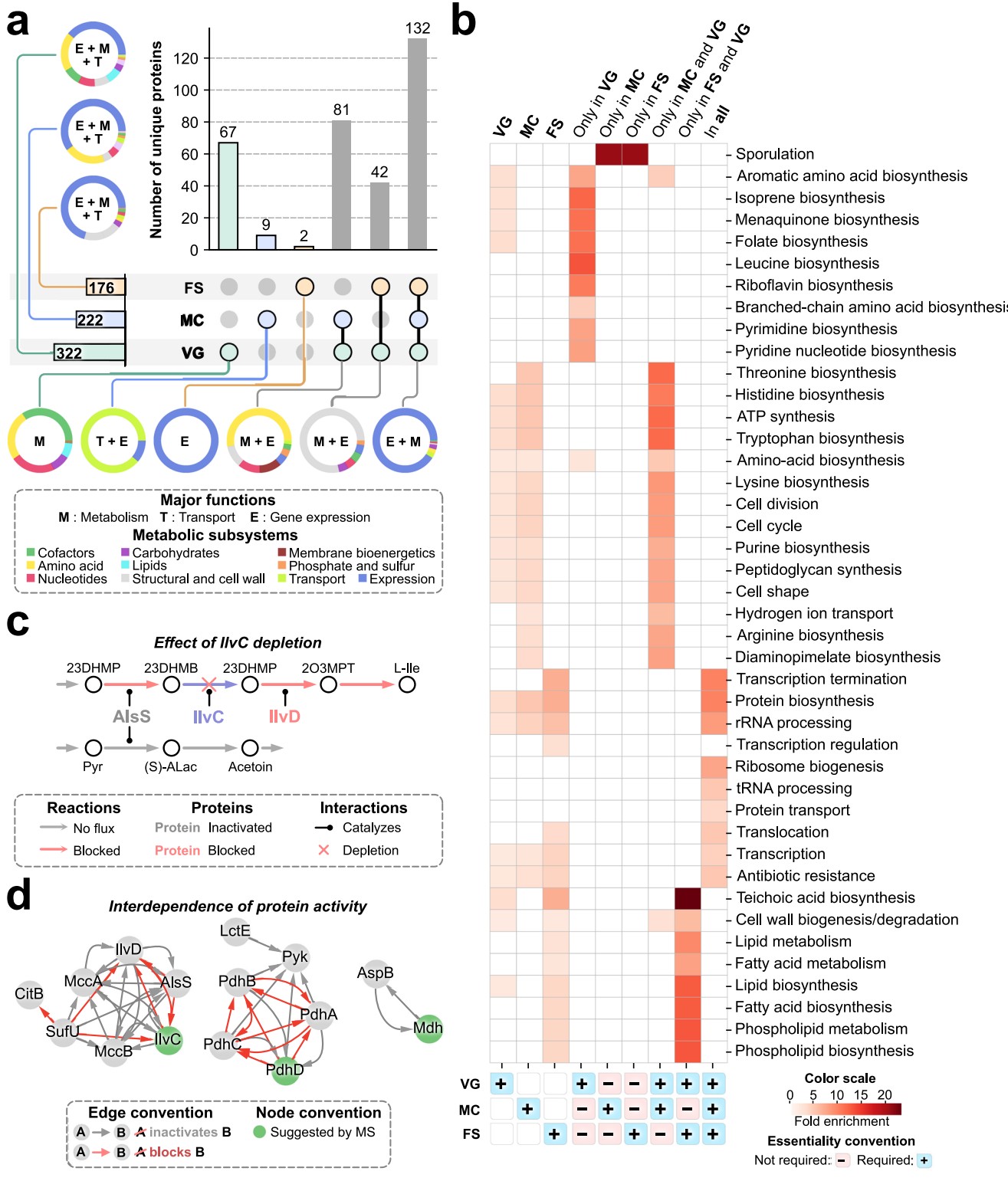

reach to be modeled due to limited information about various developmental stages. The recent discovery that developing spores metabolically differentiate[8] opened new questions on metabolism, energy production, and spore development. To answer these questions, we adapted the ME-model *i*JT964-ME with sporulation-specific constraints to generate SporeME2, a two-cell model of *B. subtilis* forespore formation. Our approach was similar to co-culture[33,34] or community modeling[18], which aim to elucidate metabolite exchange between cells. In addition, our model included the complex regulation at the transcriptional, translational, proteome, and metabolic

levels that drive the metabolic differentiation of the mother cell and the forespore[8].

For years, metabolism has not been a main focus in the *Bacillus* sporulation field. In 1968, Kornberg et al.[37] compared the enzyme composition of *Bacillus* spores to vegetative cells and found many enzymes missing or in low levels in the forespore. For decades, mechanisms derived from those absent or diluted enzymes were minimally studied, in part due to the lack of appropriate tools[9]. More recently, two proteomics studies compared the proteome makeup of the spore with vegetative cells[14,15], and one study interrogated the

**Fig. 4 | Protein essentiality and interdependence analysis in the forespore.**
**a** Upset plot summarizing the simulation results of protein essentiality. Pie charts show the partitioning of subsystems across different groups. Essentiality in the vegetative cell was assessed using cell growth rate as a proxy for growth, while forespore and mother cell used forespore formation rate instead (see Online "Methods"). **b** Pathway enrichment analysis for predicted required proteins in the forespore (FS), mother cell (MC), and vegetative cell (VG) and their intersections. Plus signs indicate requirement in FS, MC, or VG, while minus signs indicate that the protein is not required in the highlighted cell. The heatmap shows the fold enrichment of the functional annotations of the predicted essential proteins as predicted by DAVID[46]. GO enrichment analysis results from DAVID are provided in Supplementary Data 4. **c** Example of induction of inactivity and blockage as a result of IlvC depletion. Reactions are color-coded to show whether they have no flux (gray) or are blocked (red). Proteins are color-coded, gray if they are inactivated due to a depletion and red if they are blocked. IlvA and IlvD are blocked as they are involved solely in the blocked pathway, while alsS is only inactivated, as it can still be used in a second independent pathway that is predicted inactive. **d** Predicted inactivations and blockages of depletions suggested by MS. The directed graph summarizes the predicted induction of inactivity (gray arrow) or blockage (red arrow) as a result of observed depletions. The nodes represent proteins, and the arrows represent the induction of inactivity (gray arrow) or blockage (red arrow). Nodes are color-coded to show whether the protein has been suggested as a depletion by MS (PXD051727[21]). Source data are provided in the Source Data file.

metabolic differentiation of the forespore and the mother cell using a cell-specific targeted technique[8]. These studies documented major changes in the metabolic capabilities of spores, but the global impact of these changes on metabolism in the two cells has not yet been explored. The integration of our next-generation ME-model of a sporulating cell with in vivo interrogation through STRP, proteomics, and cell-specific protein synthesis assays, provides the necessary tool to study the distinct metabolism in the mother cell and the forespore. Through this approach we revealed detailed metabolic mechanisms for energy production and nucleotide synthesis. Furthermore, our study provides a potential mechanism for how sporangia shut down metabolism to prepare for dormancy.

One long-standing question is how the forespore produces the energy needed to complete sporulation. Our in vivo and in silico studies suggest that the enzymes required to produce GTP, CTP, and UTP are only required in the mother cell for spore maturation, consistent with a model that NTPs are transported to the forespore via the Q-A channel. While it is likely that ATP is also transported through the Q-A channel, it remains unclear if the Q-A channel remains open throughout sporulation. We therefore modeled what would happen in the absence of ATP transport and the model suggested that ATP can be provided to the sporangia by shuttling high energy intermediates from the mother cell into the forespore. The forespore runs glycolysis forward to produce low-energy products such as pyruvate that the mother cell consumes by performing glycolysis in reverse to produce high energy metabolites. Our GFP-tagging results support this hypothesis, but further in vivo studies are required to determine if this process occurs during sporulation and to test if the Q-A channel remains open throughout sporulation. This proposed pathway is similar to the Cori cycle in humans, where glycolytic end products are cycled from the muscles to the liver, reduced, then cycled back to the muscles to produce energy, leaving the metabolic burden of ATP production in the liver instead of the muscles[38]. Our findings suggest that the sporulation ATP cycle could be performing the same function, placing the metabolic burden of ATP production on the mother cell as the forespore transitions to dormancy.

The ME2-model allowed us to predict protein activity and essentiality during sporulation, identifying metabolic priorities and key nodes in the metabolic network. We found that the mother cell focuses on energy and amino acid production, while the forespore prioritizes the biosynthesis of structural components, likely due to its ongoing transition to dormancy. Furthermore, our essentiality analysis showed that these biomass precursor biosynthetic pathways are not only inactive, but also essential only in the MC, which further supports its nurturing role. Previous proteomics studies showed a significant reduction in amino acid and nucleotide biosynthetic proteins in the spore[14,15], which is consistent with our simulations and in vivo data. However, some proteins involved in amino acid biosynthesis were identified in the spore in these studies[14,15], which could be explained by them being required for spore germination rather than forespore formation.

Our study also identified key enzymes whose metabolic flux is shut down by depletions, and many of these predictions still remain to be confirmed. These enzymes could be targets for the forespore-specific proteolytic machinery since their degradation would shut down flux through other pathways, allowing the "unemployed" enzymes to be degraded by the Clp proteases of *B. subtilis*, which have been shown to degrade non-functional "unemployed" enzymes during starvation[39]. Combining the ME2-model with in vivo testing and cell-specific depletion via STRP provides a robust method to study spore development and cell differentiation and will allow further refinement of the model. We envision that this approach will support future studies that further define the metabolic exchanges that accompany metabolic differentiation during *B. subtilis* sporulation and will enable detailed elucidation of cell differentiation in other organisms.

## Methods

### Model reconstruction and assumptions

Our ME2-model of *B. subtilis* sporulation, *SporeME2*, was based on a recently published ME-model of *B. subtilis*, iJT964-ME. As a result, our ME2-model follows all the assumptions and formulations of COBRAme-based ME-models[17]. Furthermore, it includes metabolism and gene expression functions of *B. subtilis*, including transcription unit sigma factor specificity, as supported by BioCyc[40], UniProt[41], and SubtiWiki[42]. *SporeME2* consists of two individual ME-models representing (I) the mother cell, with no model modifications, and (II) the forespore, with 13 known protein depletions (Fig. 1c) and the absence of ATP synthase due to the lack of information supporting its presence. In vivo, protein depletions occur after sequences are translated[8,9], meaning that modeling the depletions could be done through degradation reactions of synthesized peptides. However, our model operates with the steady-state assumption, as other M- and ME-models[17,43]. This means that there is no concept of time in the simulations and that adding a degradation reaction of 100% of a specific protein pool is mathematically equivalent to not producing it at all in the first place. This is due to the optimization algorithm only choosing favorable pathways for the objective function (growth rate). Arguably, there is a cost associated with synthesizing peptides and then degrading them, which is not accounted for in our model. However, our model aims to assess the effect of not having that protein available to carry out a specific reaction, which can be assumed to be much greater than its enzymatic degradation cost. That said, all depletions were simulated in the model by closing (setting upper and lower bounds to zero) their respective translation reactions (e.g., translation_BSU00001).

We then combined the mother cell and forespore ME-models by creating separate compartments and adding all forespore and mother cell metabolites and reactions to the ME2-model in their respective compartments. This resulted in a total of three compartments in the model: the mother cell's cytosol (c), the forespore's cytosol (s), and the extracellular environment (e). When generating multi-cell models, two main questions arise: (I) How do you define the allowed metabolic exchange? and (II) How do you solve for more than one growth rate[18,44] ? Regarding the first question, metabolic

transport was allowed between the forespore's and mother cell's cytosols following the original transport reactions of *i*JT964-ME between the vegetative cell's cytosol and the extracellular environment. All transport reactions were left open following an "open pore" model where there is no clear restriction of metabolites exchanged[8].

All transport reactions were implemented from transporters in the ME-model of *B. subtilis*, *i*JT964-ME[20]. We then allowed transport via the sporulation-specific SpoIIQ-SpoIIIA complex (Q-A)[10,11,23,24]. We kept the transporter stoichiometries from *i*JT964-ME and replaced the catalyzing complex with the Q-A complex. It is worth noting that only those intermediates transported in *i*JT964-ME were allowed to be transported by Q-A in SporeME2. Previous studies have demonstrated that the Q-A channel allows calcein to move across the forespore membranes, suggesting that it is a non-specific channel[8]. Q-A assembly was reconstructed in SporeME2 according to the described σF-mediated expression of the *spoIIIAA-AH* operon in the mother cell and the σF-mediated expression of *spoIIQ* in the forespore[13]. The composition of the Q-A was reconstructed from previous reports[10,11,23,24]. Simulations were then used to identify the optimal routes that *SporeME2* chose to maximize the forespore formation rate.

Regarding the second question, our model only optimizes for the formation rate of the forespore, which is assumed to be the only objective function. However, biomass production was allowed (with open reaction bounds) in the mother cell to allow for enzymatic machinery to be synthesized and keep the steady state. Therefore, *SporeME2* was defined based on only the growth rate of the forespore (forespore formation rate). This simplification was critical in solving SporeME2, as it reduces the two-dimensional non-linear programming problem (with two variable growth rates) to a one-dimensional non-linear programming problem, which can be solved using the bisection method described in the ME-model solver package, solveME[45].

### Sigma factor specificity of gene expression in *SporeME2*
Sigma factor-specific regulation was readily incorporated in *SporeME2* by inheriting the transcription reactions from *i*JT964-ME, which contain fully annotated transcriptional units from BsubCyc. After merging forespore-ME and mother cell-ME, only one essential σK-regulated pathway remained in the forespore, peptidoglycan synthesis. According to reports, peptidoglycan is synthesized in the mother cell and incorporated in the forespore, so we allowed this membrane component to be integrated from the mother cell's cytosol. As a result, our model formulation led to σE and σK expression exclusively in the mother cell and σF expression exclusively in the forespore. σG has no annotated metabolic role relevant to our metabolic and gene expression network, so it was not included.

### Simulating flux distributions
As a consequence of being a COBRAme ME-model[17], the model optimization yields a Non-Linear Programming (NLP) problem. We used the SOLVEme[45] package to calculate flux distributions. SOLVEme iteratively assesses problem feasibility at different growth rates by substituting an assumed growth rate and solving the resulting LP problem (Eq. (1)), and uses binary search to determine the highest possible growth rate. Solving is performed through the quad-precision solver QuadMINOS 5.6.

$$\max \mu, s.t. \, S\upsilon = 0, \upsilon^L <= \upsilon <= \upsilon^U \qquad (1)$$

Where $\mu$ is the growth rate, $S$ is the stoichiometric matrix, and $\upsilon$ is a vector containing reaction flux rates.

The simulation conditions were set to resemble the experimental sporulation conditions (see Culture Conditions). Thus, we simulated a minimal medium composed of salts and supplemented with glutamate, with a lower bound of -2 mmol/gDW/h. This bound was set to a comparable value to the original glucose uptake bound in the M-model

of *B. subtilis*, -1.7 mmol/gDW/h in *i*YO844, which allows for a typical growth rate of 0.1 1/h[27]. We allowed the model to uptake several metal ions provided in the medium. The full definition of the medium used is provided in Supplementary Data 1.

### Prediction of protein essentiality
ME-models explicitly represent gene expression, which renders the definition of a protein depletion as simple as closing its respective translation reaction (see Model reconstruction and assumptions). This automatically renders the protein unavailable for any complex that requires it, and subsequently requires alternative enzymes to be synthesized, if any. A protein was deemed essential in the vegetative cell (*i*JT964-ME) if closing its translation reaction rendered the model infeasible, meaning no solution could be found. Similarly, a protein was deemed essential for forespore formation in the forespore or the mother cell if closing its translation reaction rendered the sporulation model (*SporeME2*) infeasible. We deemed a protein essential for germination and outgrowth if it is essential for vegetative growth, as it can be assumed that a germinating spore requires at least the proteins that are essential for the vegetative cell to develop and ultimately grow.

### Pathway enrichment analysis
Essential protein lists from Supplementary Data 3 were processed by DAVID[46] using the complete list of proteins in SporeME2 as a reference database. We used the functional annotations from UniProt, marked as UP_KW_BIOLOGICAL_PROCESS. The rows of the heatmap were clustered in Python 3.7 using the seaborn 0.12.0 function clustermap.

### Prediction of induction of protein inactivity and blockage
We here define blockage as the elimination of flux through an enzyme due to the accumulation of products for upstream steps in a pathway, due to an absence of precursors for subsequent steps in the metabolic pathway, or due to the proteins being part of a multi-subunit enzymatic complex. Inactivation is defined as occurring when an enzyme participates in multiple pathways, with flux through one pathway blocked by the absence of precursors and no flux through any other reaction that it could catalyze (Fig. 4c). For example, IlvC is a necessary enzyme for the conversion of threonine to isoleucine, so its depletion inactivates the entire isoleucine biosynthesis pathway. While IlvD is subsequently blocked, AlsS is only inactivated, as it can catalyze flux in a second (already predicted to be inactive) independent pathway. Although inactivation and blockage are conceptually different, both could inform of new depletions. Predicted blockages may have a higher chance of being confirmed as depletions in vivo, as they are a result of metabolic pathways rendered infeasible.

This analysis was performed to contextualize known depletions and predict possible new depletions in the forespore that have not been observed. First, given a protein A, $p_A$, we define its activity as its translation flux rate, $t_A$, so that $p_A$ is active if $t_A > 0$, and inactive if $t_A = 0$. A protein $p_A$ was deemed dependent on a protein B, $p_B$, if $p_A$ becomes inactive as a consequence of the depletion of $p_B$. It is worth noting that for the model to predict the dependence of $p_A$ on other genes, $p_A$ must be active in the wild type simulation. Therefore, $p_A$ must be allowed to be translated. For the case of constrained depletions, we reincorporated their expression, thus creating a naïve model that contained no known information on protein depletions. We then assessed protein activity before and after single depletions of all proteins in the mother cell and the forespore.

In a second in silico experiment, we aimed to confirm whether induction of inactivity was predicted due to a change in the optimal phenotype ($p_A$ is no longer favorable after $p_B$ depletion), or due to a blockage in the metabolic pathway. To test this, we added sink reactions ($\varnothing \to p_A$) for all proteins that shared interdependence with validated or GFP-observed depletions, and set the LP objective function to $\max \mu + \sum S_i$, where $S_i$ is the sink reaction rate of $i$. The

upper bound of these sinks was set to ten times the tolerance of the solver, qMINOS, $10^{-15}$, which is low enough to ensure the model is able to get flux through all sinks, but high enough to keep solution robustness[45,47]. The optimization will maximize flux through all protein sinks, and a protein will be blocked if its sink is predicted to be zero. Thus, $p_A$ was deemed blocked by depletion of $p_B$, if $S_A$ becomes zero. Naturally, a predicted blockage must also be a predicted inactivation, which is the case for all blockages shown in Fig. 4.

## Strain construction

All strains used in this study are derivatives of *Bacillus subtilis* PY79. The supplemental information includes a list of strains (Supplementary Table 2), plasmids (Supplementary Table 3, Supplementary Methods), and oligonucleotides (Supplementary Table 4), as well as detailed descriptions of strain construction. All *B. subtilis* strains were constructed by transformation using genomic DNA and a 2-step competence protocol unless otherwise noted. Plasmid integrations were confirmed by PCR. Antibiotic concentrations used for selection after transformation of *B. subtilis*: 10 μg/mL kanamycin, 100 μg/mL spectinomycin, 5 μg/mL chloramphenicol, 10 μg/mL tetracycline, 1 μg/mL erythromycin and 25 μg/mL lincomycin.

## Culture conditions

*B. subtilis* strains were generally grown on LB agar plates at 30 °C for culturing. Sporulation was induced via resuspension. For fluorescence microscopy experiments, cells were grown to O.D.600 ~ 0.6 in ¼ diluted LB, and sporulation was induced by resuspension in A + B sporulation medium containing glutamate as the sole carbon source[48]. The induction of sporulation was considered to be the moment at which the cells were resuspended in A + B medium. Sporulation cultures were grown at 37 °C for batch culture experiments, spore titers, and spore purification, and at 30 °C for timelapse microscopy experiments.

## Batch culture microscopy

Microscopy was performed as described in Riley et al.[8]. Cells were visualized on an Applied Precision DV Elite optical sectioning microscope equipped with a Photometrics CoolSNAP-HQ2 camera. Images were deconvolved using SoftWoRx v5.5.1 (Applied Precision). The median focal planes are shown. To assess the GFP fluorescence signals in the mother cell and forespore, sporulation was induced by resuspension in A + B medium. To visualize the membranes, 0.5 μg/mL FM 4-64 was added to the culture 1 hour after sporulation induction, and sporulation was allowed to proceed for an additional 2 hours under standard culturing conditions. At three hours following sporulation induction, 15 μl of culture was transferred to 1.2% agarose pads, prepared in A + B medium and supplemented with an additional 0.5 μg mL −1 FM 4−64 (Life Technologies). Excitation/emission filters were TRITC/CY5 for membrane imaging (100 ms exposure time) and FITC/FITC for GFP imaging (600 ms exposure time), with excitation light transmission set to 100% for both filters.

We also used microscopy to assess the completion of two developmental milestones: engulfment completion and forespore maturation. Engulfment completion was monitored using a well-characterized membrane fission assay[26]. Briefly, cells are treated with a red membrane-impermeable membrane dye, FM 4-64, and a green membrane-permeable membrane dye, MitoTracker Green (MTG). During engulfment, forespore membranes are accessible to both dyes. Following membrane fission, however, the forespore membranes can no longer be stained by the red dye and are therefore only labeled by the green dye. Forespore maturation was assessed by phase contrast microscopy. Mature spores become partially dehydrated, which confers upon them a bright appearance under phase-contrast microscopy. Thus, the extent to which developing spores become phase-bright can be used as a metric of spore maturation.

## Forespore to mother cell GFP fluorescence ratio quantification

We used the forespore:mother cell fluorescence ratio as a measure of the extent to which forespore enzymes were depleted from the forespore since the fluorescence signal in the mother cell remained more or less constant throughout engulfment for each GFP fusion protein. This allowed us to correct for any effects caused by photobleaching, and also allowed us to more readily compare across fusions, given the variability in the abundance of different proteins. The GFP intensity in the forespore and mother cell was measured using a custom Matlab 2017b script as described in detail in Riley et al.[8]. Measurements for each strain were taken from individual cells in the same sample.

## Spore titer assay

Sporulation was induced in 10 ml of A + B resuspension medium and was allowed to proceed at 37 °C for 24 hours. Two milliliters of culture was then heated at 80 °C for 20 min, serially diluted in 1× T-base, plated on LB, and incubated overnight at 30 °C. Spore titers were calculated based on colony counts. Measurements were taken from distinct samples.

## Spore purification

Sporulation was induced in DSM and allowed to proceed for 72 hours at 37 °C. Cultures were pelleted, washed once with 4 °C sterile water, and incubated overnight at 4 °C in sterile water to lyse the remaining vegetative cells. The next day, the spore samples were pelleted, washed once with 4 °C sterile water, incubated overnight at 4 °C in sterile water and purified over a phosphate-polyethylene glycol aqueous biphasic gradient as previously described[48], harvesting spores from the organic phase, and washing with 50 or more volumes of 4 °C sterile water at least three times. The purified spores were resuspended in fresh sterile water and further purified using a histodenz step gradient. Sample purity was evaluated using phase-contrast microscopy. Purified spores were stored in sterile water at 4 °C.

## Monitoring germination by phase-contrast timelapse microscopy

Purified spores were diluted to an O.D.600 of 0.3 in sterile water, and 10 μL of the spore suspension was applied to 1.2% agarose pads prepared in LB and supplemented with the 10 mM of the germinant, L-alanine. Pads were partially dried, covered with a glass cover slip, and sealed with petroleum jelly to avoid dehydration during time-lapse imaging. Phase-contrast imaging was performed in an environmental chamber set to 30 °C. Images were acquired every 3 min for 10 hours using POL/POL filters. Light transmission was set to 32% and exposure time was 0.1 s. Note that, due to sample preparation, there was a time lag of ~15 min before imaging commenced. To minimize the germination of spores on the pads during this time, spores were not heat-activated before performing timelapse microscopy. We focused on spores that were phase-bright at the onset of imaging. Images for each strain were taken from the same sample measured repeatedly.

## Reporting summary

Further information on research design is available in the Nature Portfolio Reporting Summary linked to this article.

# Data availability

The reconstructed ME2-model of *Bacillus subtilis* sporulation, SporeME2, along with all scripts, code, and data used in this manuscript, have been deposited at https://github.com/jdtibochab/sporeme. The Supplementary Data has been deposited at Figshare under the accession code https://doi.org/10.6084/m9.figshare.25800394. Source data are provided with this paper.

## Code availability

The code used for all analyzes and reproducing all figures is available at https://github.com/jdtibochab/sporeme[49].

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

## Acknowledgements

This work was supported by the U.S. Department of Energy, Office of Science, Office of Biological and Environmental Research, and Genomic Science Program under Secure Biosystems Design Science Focus Area (SFA) contract number DE-AC36-08GO28308, by the National Science Foundation under award numbers EFMA-2223669 and DMS-2325172, and by the NIH Grant R01-GM57045 (to K.P.). J.L. was supported by the PiBS NIH T32 Grant (GM133351) and the UC-HBCU Initiative. J.T.-B would like to thank Cristal Zuniga for fruitful discussions and guidance. We thank Javier Lopez-Garrido and Alan Derman for their helpful discussion.

## Author contributions

Conception and design of the study: J.T.-B., K.Z., J.L., E.P.R., K.P., K.Z.; acquisition, analysis or interpretation of the data: all authors; development of metabolic model: J.T.-B., K.Z.; writing of the manuscript: J.T.-B., J.L., K.Z., K.P.

## Competing interests

The authors declare that there are no competing interests.
