## [Transparent Peer Review file · Nature Communications]

Deciphering metabolic differentiation during *Bacillus subtilis* sporulation

Corresponding Author: Professor Karsten Zengler

Version 0:

Reviewer comments:

Reviewer #1

(Remarks to the Author)

Tibocha-Bonilla and colleagues present an interesting manuscript addressing the metabolic differentiation of *Bacillus subtilis* during sporulation through a multidisciplinary approach. First, a global ME model of the mother cell (MC) and the forespore (FS) is constructed. Then, the model is used to assess the role of key metabolic hubs in cellular differentiation, including nucleotide, energy, and amino acid provision. Some model predictions are elegantly validated using STRP and phase microscopy. Finally, the model is used to predict metabolic reprogramming in the FS through in silico gene essentiality analysis.

The approach has novelty in the field, the manuscript is valuable and includes a substantial workload; however, some sections require further contextualization to be fully exploited and to enhance the current scope of the study. Additionally, greater accessibility to the constructed model is necessary for the research community.

1. While the nucleotides and energy provision is consistent between model and experimental data, the provision of amino acid to FS requires additional clarification, validation, and discussion.

Model predictions suggest minimal intervention of the MC in supplying amino acids to the FS. According to supplementary Data 2, the percentage of total production of the most of amino acids in the FS is 100%. However, these predictions are significantly at odds with the experimental data, which show notable protein dilution for key enzymes in amino acid biosynthetic pathways, including MetE, ArgD, and ArgB. Furthermore, the substantial dilution observed for IlvE indicates that de novo synthesis of branched-chain amino acids is also inhibited in the FS. Which agree with authors conclusions in discussion.

Taken together, the simulation results, validation experiments, and further discussions do not fully explain how amino acid provision is managed in the FS, and how the model captures this aspect, if it does. In any case, the major claim by the authors regarding the energetics as model-driven force for amino acid provision is far from being demonstrated in the manuscript.

Could please the authors clarify this point.

2 Regarding gene essentiality.

-This is a very nice analysis, but it only provides a descriptive overview of model predictions. It would be interesting to contextualize this model-guided analysis within the experimental data. Overall, this analysis introduces some uncertainty in the manuscript, as it does not fully align with previous findings. For instance, nucleotide, sugar phosphate, and amino acid transport appear to be non-essential for the FS. Is this because these three important metabolic hubs are highly robust by design to ensure sporulation even under perturbations? Or is this because the FC ME model lacks additional constraints in terms of proteins dilution. Could the author provide any insight on this?

-Would it be possible to perform a synthetic lethality analysis? This approach would help identify potential vulnerabilities in the FS that are not apparent under single perturbation conditions.

-Additionally, can the authors contextualize the protein essentiality data with the previous sections of the manuscript and with the available experimental data?

3. On the other hands, the author predicted new protein depletions that have not yet been validated, which could significantly

impact FS metabolism. Validating such predictions would be interesting and could provide additional insights into amino acid metabolism.

4. Related to the code used to generate the model, I encountered difficulties running it according to the provided instructions. To ensure reproducibility of the results, it would be needed to clarify and complete these instructions or resolve the existing issues. For example, I found local folder references, likely from the authors' computers, and the software requires the COBRAme package to be pre-installed, which was not specified. Improving the accessibility and usability of the code is crucial for the broader research community to validate and build upon this work. Please provide a more user friendly and accessible version of the code.

5. Furthermore, I couldn't find the final model, including the mother cell and the forespore, in the provided files. They might be among the large number of files in the GitHub repository. I suggest creating a direct link in the README or a specific folder for the final model to facilitate its identification. This is particularly important because the model represents one of the main advances of this publication, and having easy access to it would greatly benefit the readers and researchers who wish to utilize and build upon your work.

(Remarks on code availability)

Related to the code used to generate the model, I encountered difficulties running it according to the provided instructions. To ensure reproducibility of the results, it would be needed to clarify and complete these instructions or resolve the existing issues. For example, I found local folder references, likely from the authors' computers, and the software requires the COBRAme package to be pre-installed, which was not specified. Improving the accessibility and usability of the code is crucial for the broader research community to validate and build upon this work. Please provide a more user friendly and accessible version of the code. Furthermore, I couldn't find the final model, including the mother cell and the forespore, in the provided files. They might be among the large number of files in the GitHub repository. I suggest creating a direct link in the README or a specific folder for the final model to facilitate its identification. This is particularly important because the model represents one of the main advances of this publication, and having easy access to it would greatly benefit the readers and researchers who wish to utilize and build upon your work.

Reviewer #2

(Remarks to the Author)

(Remarks on code availability)

Reviewer #3

(Remarks to the Author)

The authors sought to address the long-standing question of how the forespore produces the energy needed to complete sporulation.

Using the model prediction approach, the authors identified 1,707 proteins, 11,389 reactions, and 7,440 metabolites both in both the mother cell and forespore compartments. They proposed that the sporulation-specific SpoIIQ-SpoIIIA complex (Q-A) transports metabolites from the mother cell to the forespore.

Using several of the C-terminal tagged GFP enzymes (~16), the authors experimentally demonstrated compartment specific localization of the tested proteins.

STRP-mediated depletion of enzymes in the mother cell or forespore was used to assess the essentiality of enzymes involved in the synthesis of NDPs or NTPs in one or both cells. The enzymes analyzed were PyrG (CTP synthase), Gmk (guanylate kinase), and Cmk (cytidylate kinase).

Strains that degrade PyrG, Gmk, or Cmk in the forespore are able to produce spores but do not outgrow, suggesting that these enzymes are essential for vegetative growth.

The results indicated that degradation of PyrG, Gmk, or Cmk is dispensable in the forespore, suggesting that NDPs or NTPs synthesized in the mother cell are transported to the forespore through the Q-A channel or an unidentified channel.

The results suggest that the enzymes required to produce GTP, CTP, and UTP are only required only in the mother cell for spore maturation, consistent with a model in which NTPs are transported to the forespore via the Q-A channel or an unidentified channel.

They proposed that if the Q-A channel is inactivated after engulfment, energy (ATP) could be produced in the forespore by glycolytic enzymes. Thus, the authors concluded that glycolytic enzymes transfer ATP to the forespore.

The authors interpreted their results by suggesting that if the Q-A channel is inactivated after engulfment, ATP could be produced in the forespore using glycolytic enzymes, including Pyk.

Specific questions and concerns.

The synthesis of housekeeping factors, including core subunits of RNA polymerase and ribosome subunits, and essential metabolites, such as NTPs and amino acids, is mainly controlled by the housekeeping SigmaA RNA polymerase, which is generally known to be inactivated during sporulation when the sporulation-specific Sigma factors (SigmaE, K, F and G) become active (PMID: 10200951, 7708009, 4198276, Subtiwiki website). NTPs, including ATP and GTP, amino acids, RNA polymerase core enzyme, and ribosome subunits are essential for the synthesis of mRNAs and proteins in the forespore. Therefore, the available data suggest that most of these essential housekeeping components are synthesized prior to polar septum formation and that initial concentrations of these components distributed in the sporangium are sufficient to complete sporulation. Thus, it is not clear whether the authors' model of "the forespore producing the energy needed to complete sporulation" is correct. Overall, I feel that the model proposed here is not necessarily supported by the published and available data. The authors need to address these uncertainties.

<https://pubmed.ncbi.nlm.nih.gov/4198276/>

While SspB is expressed specifically in the mother cell and the forespore, the subunits of ClpXP are expressed under the control of SigmaA RNA polymerase. Again, SigmaA is inactivated in the forespore compartment. Therefore, the subunits of ClpXP appear to be synthesized before polar septation and to be distributed to the forespore compartment. As a result, the protease activity remains in the forespore compartment after polar septation. Based on these observations and facts, the housekeeping factors are not necessarily synthesized in the forespore compartment. The authors need to investigate this paradox.

Glycolytic genes, including pfkA, fbaA, tpiA, gapA, pgk, pgm, eno, and pyk, are not essential (PMID: 23420519). These results suggest that, under certain conditions, ATP synthesis may be achieved by other pathways, not primarily by glycolytic enzymes. This possibility needs to be investigated by the authors.

The gapA gene in the hexacistronic cggR gapA pgk tpi pgm eno operon (the gapA operon) is repressed by CggR, the product of the first gene in the absence of glucose or other sugars and is expressed only in the presence of glucose (PMID: 11489127, 10799476). These results do not support the data presented by the authors in this study (Fig. 3). The authors need to comment on this issue.

"Amino acid supply to the forespore is driven by energetics"

"Protein essentiality predictions reveal metabolic reprogramming at the proteome level"

"Prediction of new possible proteome-scale depletions".

These topics have only been predicted by modeling. Authors must provide experimental data to support these predictions.

Intracellular metabolite concentrations have been determined experimentally in E. coli (PMID: 19561621). Can you estimate the concentrations of key metabolites and the energy requirements for cell division and spore development? The authors need to investigate these points to estimate the concentrations of the metabolites in B. subtilis.

References for PyrG CTP synthase (essential, SigA controlled), Gmk guanylate kinase (GTP biosynthesis, essential, SigF controlled), and Cmk cytidylate kinase (cytidylate kinase, essential) need to be added for results and discussion, such as expression before and after polar septation.

The authors describe: "Together, these results suggest that the Q-A channel, or an unidentified channel, transports NMPs, NDPs, or NTPs to the forespore during spore formation". Is it possible that there is an unidentified channel that transports NTPs? An unidentified channel would be essential for sporulation if it existed, but such "sporulation" genes have never been identified. The authors need to clarify this.

The channel(s) would be a passive transport system. Please discuss this to exclude the possibility of active transport or any other types of transport.

Experimental data on genes controlled by compartment specific sigma factors have been published (PMID: 10869437, 12169614, 16497325, 15383836, 12662922). The authors must mention how these published data support the authors' model prediction, such as the expression of glycolytic enzymes in the forespore compartment during sporulation.

In Fig. 3C, how were these 16 proteins selected from many others for the GFP-tagged experiments. The imaging data would be helpful to verify the data in Fig. 3C.

(Remarks on code availability)

(Remarks to the Author)

Dear authors,

I have read with great interest your manuscript which I think nicely highlights the interdisciplinary nature of the work where modelling, spatiotemporal-regulated proteolysis (STRP) and GFP-tagging meet once more. The 2021 Science Advances paper was in that sense more ground-breaking as it is a fully new view on the cellular differentiation at the molecular metabolic level. The current paper is in a way a specialized further detailing of the approach pioneered in the Science Advances paper. In fact, the current focus is on the Q-A channel and its role in energy metabolism of the fore spore as well as on shuttles and proteins putatively involved in these. While the enzymes involved in gluconeogenesis and glycolysis partition as can be logically explained from a mother cell forespore nurturing perspective, expected the presumed membrane transporter proteins have not (yet) been unequivocally identified. Can the authors comment on/ discuss this. Additionally, the amino acid and protein biosynthesis pathways are modelled and analyzed. The logic of protein biosynthesis in the fore spore and amino acid production supplying the fore spore is compelling and illustrated nicely with the experiments done.

The overall focus on metabolism as a key element in understanding cellular differentiation is laudable and unique. It is remarkable though that the authors do not put their findings in perspective of the proteomics and metabolomics data on spore and vegetative cell composition in *Bacillus* as published in *mSphere* in 2020 and this year in *J. Proteome Res.* (Huang et al. 2024). This omission should be adjusted.

Attached to this review the annotated manuscript contains still some specific comments to address.

(Remarks on code availability)

Reviewer #5

(Remarks to the Author)

Deciphering metabolic differentiation during *Bacillus subtilis* sporulation

Tibocha-Bonilla, Lyda et al. follow an *in silico* and *in vivo* approach to study sporulation in *Bacillus subtilis*. Sporulation involves an asymmetric cell division producing two distinct cells - a large mother cell and a small forespore; each tightly regulated by gene expression. A recent study hypothesized that the forespore is highly dependent on the mother cell, since the forespore needs building blocks produced by the mother cell for protein synthesis required to complete spore assembly. Motivated by the latter hypothesis, the authors present a novel approach to study cross-feeding interactions between mother cell and forespore. For that, they extrapolate the principles of community modeling framework used with metabolic models (M-models/GEMs) to model cell differentiation by using a community metabolic and gene expression (ME-) model of the two cell types. They construct a community ME-model (SporeME2) made by two ME-models, one representing the mother cell and one representing the forespore, allowing metabolite cross-feeding between the two cells.

Model predictions were then tested by *in vivo* analyses making the study more robust. Predicted metabolic interactions between the two cells were used to guide spatiotemporal regulated proteolysis (SPTR) and C-terminal GFP-tagging experiments and further fluorescence microscopy. Following this approach the authors gain new insights into the mechanisms of spore assembly and cell differentiation in *B. subtilis*. Furthermore, the presented findings can help expanding the understanding of the same mechanisms in other organisms.

While the study is innovative and present valuable insights and contributions to the field, the authors need to clearly explain, develop and justify the methodology used to facilitate the understanding, validation and reproducibility of their modeling results. Below there is a list of comments that the authors need to address to achieve that.

Major comments

1. Methods. The authors indicate that they use a similar approach to community modeling. In a community model (M-models), each species has their own biomass reaction, and each species is represented as a single compartment. One can then create a community biomass reaction by adding the weighted contribution of the biomass of each species (using their respective metabolite), and set this reaction (if applicable) as the objective function. Many studies and tools have shown that integrating the relative abundance/biomass species ratio within the community modeling framework is key to correctly applying this approach (<https://journals.plos.org/ploscompbiol/article?id=10.1371/journal.pcbi.1005539>, <https://journals.asm.org/doi/10.1128/msystems.00606-19>, <https://journals.plos.org/ploscompbiol/article?id=10.1371/journal.pcbi.1011363>), as it might also lead to different flux distributions.

While the framework used in e.g. SteadyCom is not compatible to ME-models, due to the non-linearity issue, one can constrain the growth rates and scale the exchange fluxes by the strain/cell relative's abundance (<https://www.sciencedirect.com/science/article/pii/S2001037020304256>). In fact, this methodology has been implemented already in a community model using ME-models (<https://journals.plos.org/ploscompbiol/article?id=10.1371/journal.pcbi.1006213>).

Considering the authors are using a community model to study the interactions between the two cells, and the two cells differentiate in volume, and thus, in biomass, the authors need to clarify how they account (if they do) for the relative abundance in their modeling framework. If the authors do not account for the relative abundance, they need to justify their methodology used to validate their approach.

2. Following the previous comment, the authors seem to tackle that matter by only considering growth of the forespore in SporeME2 model. The code shows a spore biomass reaction and a section that mentions 'Mother is not growing' (BuildCommunityModel.ipynb). Considering the mother cell produces the main building blocks (biomass precursors) for the forespore, which is a sign of cell growth, it is not clear how the authors represent growth (if represented) on the mother cell. Does the mother cell have a separate biomass reaction? How is it defined and constrained in the model? The authors need to clarify this point in the text and justify their modeling framework.

3. Properties of the SporeME2 model ...

The authors mention: 'Supplemental Data 1 contains the flux predictions'. However, it is not clear what are model predictions indicating here, and how are the fluxes obtained in the model, under which conditions, what are the import fluxes or how they are constrained (in methods). The authors need to clarify all this information in the text to facilitate the understanding and reproducibility of their results.

4. The installation of sporeme is not sufficiently clear in the github repository. The authors need to add the cobrame installation within the installation steps. There is a lack of information on the different versions required to run sporeme. This creates compatibility issues (e.g python, numpy, optlang..) when first running python setup.py develop -user, suggesting this is not the most appropriate method.

This could be solved by creating a virtual environment in conda with the specific versions, or using the docker image in COBRAME.

However, the main issue is with cobra. The authors recommend to use cobra 0.5.11, but this version is not available when using pip install cobra=0.5.11, and neither in bioconda. Therefore, it is crucial that the authors solve this issue upon validation of their results.

Minor comments

1. Results. Properties of the SporeME2 model ...

The authors seem to build the SporeME2 by aggregating two ME-models. The mother cell model seems to be the iJT964-ME plus transporters, and the forespore model is adapted from the original iJT964-ME by correcting for protein depletions and allowing transport through A-Q complex. This is clear in the methods and Figure 1.c, but the results could benefit of a brief explanation too.

2. The authors need to clarify in the text how are the transport of intermediates via the Q-A complex represented in the model.

3. The intracellular reactions of each species are defined in two cytosolic compartments as mentioned by the authors, and metabolites can be transported from the cytosolic compartment of the mother cell to the extracellular compartment and from the extracellular compartment to the cytosolic compartment of the forespore and vice versa. However, the cytosolic compartments need to be defined with different ids ('_c', '_s'), so can the authors briefly clarify this in the text? Are there more compartments in the model as shown in the code?

4. Results. The authors need to clarify to the reader how are model predictions indicating that spore assembly can be achieved.

5. Introduction: The authors refer to 'Recent studies' but then they only use one reference (8). Then they talk about 'The publication used ..'. The authors can cite other studies or rephrase the sentence.

6. Introduction. 'multi-cell ME-models are yet to be explored'. Since there are yet not many studies where they implement community ME-models, it is fair that the authors refer to the previous study where they implement community modeling with ME-models (<https://journals.plos.org/ploscompbiol/article?id=10.1371/journal.pcbi.1006213>).

7. The authors add a github repository with the SporeME2 model and all the scripts and data used in the study. However, one does not realize the existence of this repository till the end of the manuscript. For clarity to the reader and further use of this approach, the authors can refer to the existence of this repository in the main text.

8. Check if the reference to the Supplementary Fig. 3 is correct in the 'STRP and GFP-tagging interrogate mother cell...' section.

9. Discussion. 'Since then, mechanisms derived from those absent or diluted enzymes have been minimally studied (4), in part due to the lack of appropriate tools (7)'. Correct the format of the two references here.

10. Model reconstruction and assumptions. Correct reference 4 and 9 in this section.

(Remarks on code availability)

I ran the code in a linux machine following the installation steps. First I got the error for the cobrame not being installed. So, I read the readme file again where they mention cobrame (before the installation steps). So, the installation of cobrame (or link) and required solvers should be clearly stated in the installation steps. Once I installed cobrame I got an issue running

1.1.BuildCommunityModel.ipynb due to a compatibility issue with the numpy version. Then I checked, and the version you need is 1.15 that seems to work only until python 3.7, which is not something one realized right away. So I created a conda environment and install a downgraded version of python (3.7) and the compatible version of numpy (1.15). Then I tried to run the 1.1.BuildCommunityModel.ipynb again, and I had to install tqdm. I then got an issue with optlang. I installed it, and again one needs to check which version is compatible to run with the rest of packages. Then I got this error:

```
/anaconda3/envs/sporeme/lib/python3.7/site-packages/cobra/io/_init__.py:10: UserWarning: cobra.io.sbml requires libsbml
warn("cobra.io.sbml requires libsbml")
cobrame/_init__.py:30 UserWarning: COBRAPy version is 0.5.4. We recommend using 0.5.11. Using earlier versions may
cause errors
```

So I tried to install cobra version 0.5.11 .

However, there is no such version:

```
pip install cobra==0.5.11
```

```
ERROR: Could not find a version that satisfies the requirement cobra==0.5.11 (from versions: 0.2.0, 0.2.1, 0.3.0b1, 0.3.0b2,
0.3.0b3, 0.3.0b4, 0.3.0, 0.3.1, 0.3.2, 0.4.0a1, 0.4.0a2, 0.4.0a3, 0.4.0a4, 0.4.0b1, 0.4.0b2, 0.4.0b3, 0.4.0b4, 0.4.0b6, 0.4.0b7,
0.4.0, 0.4.1, 0.4.2b1, 0.4.2b2, 0.5.1b1.post14, 0.5.2b3, 0.5.2, 0.5.3.post3, 0.5.4, 0.8.2, 0.9.0, 0.9.1, 0.10.0a1, 0.10.0, 0.10.1,
0.11.0, 0.11.1, 0.11.2, 0.11.3, 0.12.0, 0.12.1, 0.13.0, 0.13.1, 0.13.2, 0.13.3, 0.13.4, 0.14.0, 0.14.1, 0.14.2, 0.15.0, 0.15.1a0,
0.15.1, 0.15.2, 0.15.3, 0.15.4, 0.16.0, 0.17.0, 0.17.1, 0.18.1, 0.19.0, 0.20.0, 0.21.0, 0.22.0, 0.22.1, 0.23.0, 0.24.0, 0.25.0,
0.26.0, 0.26.2, 0.26.3, 0.27.0, 0.28.0, 0.29.0)
```

```
ERROR: No matching distribution found for cobra==0.5.11
```

and neither in bioconda.

```
cobra 0.4.0b6 py27_0 bioconda
cobra 0.4.0b6 py34_0 bioconda
cobra 0.4.0b6 py35_0 bioconda
cobra 0.4.0 py27_0 bioconda
cobra 0.4.0 py27_1 bioconda
cobra 0.4.0 py34_0 bioconda
cobra 0.4.0 py35_0 bioconda
cobra 0.4.0 py35_1 bioconda
cobra 0.4.0 py36_0 bioconda
cobra 0.4.0 py36_1 bioconda
cobra 0.10.1 py27_0 bioconda
cobra 0.10.1 py35_0 bioconda
cobra 0.10.1 py36_0 bioconda
cobra 0.10.1 py_1 bioconda
cobra 0.15.4 py_0 bioconda
```

And then I stop here because I cannot run it anymore if the only recommended version to run cobra is not available anymore.

Based on this revision:

```
python setup.py develop --user
```

would not be the most appropriate command as it will install different versions that are not compatible with one another. So, all the dependencies and versions should be clearly stated.

Alternatively, COBRAME link mentions the possibility of working with a docker image, which would help to solve the compatibility issues.

However, I don't know if it would work, as it happens when creating a conda environment.

It is essential that the authors correct these issues in order to validate their method, and thus, their results. Once they do it, they need to extend and clarify the installation steps and all the dependencies required to properly run sporeme.

Version 1:

Reviewer comments:

Reviewer #1

(Remarks to the Author)

The revised manuscript by Tibocho-Bonilla and colleagues has significantly improved in clarity and understandability. Many of my previous concerns have now been addressed, but others remain, especially those related to the additional experimental validation of key predictions. Additionally, access to the model and the code is still challenging, which hampers not only reproducibility but also further use of this material by the community. Altogether, I feel that at least partial resolution of these shortcomings is needed to consider the current manuscript for publication.

1. A major conclusion of this piece is that while the MC focuses on energy and amino acid production, the FS prioritizes the biosynthesis of structural components. In this context, I found the analysis of ATP production in the FS interesting; however, I really miss a detailed analysis and discussion about the production of NADPH, the other key resource for biosynthesis. How is NADPH generated in the FS? Can the authors provide a specific analysis on this topic and include the NADPH-producing pathway in Figure 3A? I consider this aspect essential for understanding the biosynthetic metabolism predicted in the FS.

2. As I mentioned earlier, the analysis of ATP production in the FS is very interesting, as it aligns well with previous findings and is partially validated by the high levels of Pyk and GapA observed. A side effect of this predicted glycolytic metabolism is the production of NADH. The authors subsequently suggest that LctE plays a key role as the primary regulator of NADH/NAD⁺ balancing. This is indeed an interesting hypothesis that, in my opinion, requires further exploration. In my previous report, I requested validation of some new model predictions. I understand the authors' reluctance to provide a systematic validation; however, experimentally validating the role of LctE could provide solid evidence for this predicted glycolytic metabolism in the FS by focusing on a single protein. An additional STRP and GFP-tagging analysis targeting LctE would be valuable. Also a single LctE knockout could also be interesting, as it may compromise sporulation if the authors' hypothesis is correct.

3. When I attempted to install the software using Docker (which is an excellent choice for reproducibility), I consistently encountered an error related to the libsm1 library during the Docker build step. I tried this on multiple Linux systems with different OS configurations, but the same error occurred each time. I am wondering if this is a bug or if I need a specific installation setup or Docker version to resolve it. In my opinion, being able to reproduce the results is crucial.

4. The reliance on outdated and unmaintained packages (like cobra 0.5.11) poses a significant challenge for reproducing the authors' research. I recommend updating the code or at least providing an updated installation method for the required packages to ensure the code is reproducible.

5. Regarding the Jupyter notebooks for model generation, I noticed that both the "mother" and "spore" models have the same title. Is this an error? It's difficult to differentiate between them, as the code is nearly identical. If these models share a significant amount of code, I suggest creating a reusable function or package to consolidate the shared components, which would make the code easier to understand and maintain.

Overall, a more user-friendly installation pipeline and updated libraries are needed to ensure model reproducibility and facilitate further use

(Remarks on code availability)

3. When I attempted to install the software using Docker (which is an excellent choice for reproducibility), I consistently encountered an error related to the libsm1 library during the Docker build step. I tried this on multiple Linux systems with different OS configurations, but the same error occurred each time. I am wondering if this is a bug or if I need a specific installation setup or Docker version to resolve it. In my opinion, being able to reproduce the results is crucial.

4. The reliance on outdated and unmaintained packages (like cobra 0.5.11) poses a significant challenge for reproducing the authors' research. I recommend updating the code or at least providing an updated installation method for the required packages to ensure the code is reproducible.

5. Regarding the Jupyter notebooks for model generation, I noticed that both the "mother" and "spore" models have the same title. Is this an error? It's difficult to differentiate between them, as the code is nearly identical. If these models share a significant amount of code, I suggest creating a reusable function or package to consolidate the shared components, which would make the code easier to understand and maintain.

Overall, a more user-friendly installation pipeline and updated libraries are needed to ensure model reproducibility and facilitate further use

Reviewer #2

(Remarks to the Author)

(Remarks on code availability)

I couldn't install it with the docker instructions provided by the authors.

Reviewer #3

(Remarks to the Author)

I appreciate the authors' efforts to respond to my comments. All my questions and suggestions have been addressed by the authors. I have no further comments.

(Remarks on code availability)

Reviewer #4

(Remarks to the Author)

Dear authors,

I am satisfied with the answers provided and would like to commend you on the significant improvements you have made in response to the challenges associated with the GitHub repository.

The two installation methods now offered—one traditional local installation and one using Docker—are well-considered and provide a valuable balance between maintaining consistency with established protocols and ensuring ease of use. The traditional approach, while requiring several manual steps, is aligned with practices seen in other ME-model projects, and I appreciate that you have kept it for users who prefer a more hands-on approach. Your transparency regarding the necessity to manually install COBRAPy 0.5.11, despite its removal from PyPI, is especially commendable, as it ensures users are fully aware of potential challenges in advance.

The Docker-based solution is a major enhancement. By offering also this more streamlined option, you have significantly reduced the complexity of setting up the computational environment, making it much more accessible to a wider audience. The clear and well-documented instructions for creating a Docker image and running a container, along with the Jupyter Notebook session that simplifies result reproduction, demonstrate your commitment to making the research more reproducible and user-friendly.

Overall, your efforts to improve usability, enhance reproducibility, and provide clear documentation have substantially strengthened the quality and impact of this study. I appreciate the time and care you've invested in addressing reviewer feedback, and these changes undoubtedly make the work more robust and accessible to users across diverse areas of bacterial spore studies and beyond.

(Remarks on code availability)

I indeed (now) managed to get the code running.

Reviewer #5

(Remarks to the Author)

I would like to thank the authors for their careful consideration of the comments. I have reviewed the revised manuscript and the code and I am pleased to see that the authors have extensively addressed the feedback in a thorough and thoughtful manner. The revisions have strengthened the manuscript, and the clarifications/corrections provided improve both the quality of the work and the reproducibility of their results.

I have just one follow-up comment regarding the authors' response to comment 2. Once this point has been addressed, and given the positive incorporation of the rest of the feedback, I will consider the manuscript acceptable for publication.

Regarding comment 2 from the first revision:

2. Following the previous comment, the authors seem to tackle that matter by only considering growth of the forespore in SporeME2 model. The code shows a spore biomass reaction and a section that mentions 'Mother is not growing' (BuildCommunityModel.ipynb). Considering the mother cell produces the main building blocks (biomass precursors) for the forespore, which is a sign of cell growth, it is not clear how the authors represent growth (if represented) on the mother cell. Does the mother cell have a separate biomass reaction? How is it defined and constrained in the model? The authors need to clarify this point in the text and justify their modeling framework.

We thank the reviewer for catching the lack of description of the objective function and the solution strategy in the manuscript. Precisely, both cells present growth rates, as the reviewer correctly inferred, because they produce expression machinery, which must be diluted to biomass in order to keep the steady state. However, the key simplification that allowed us to solve this model is that only the forespore's growth rate is optimized, giving an optimization problem that is still non-linear but only one-dimensional. In this way, we were able to use the bisection method described previously in the solver package, solveME. We have now included a paragraph in the methods providing these details, and have also updated the notebook's misleading details. The paragraph reads:

“Regarding the second question, our model only optimizes for the formation rate of the forespore, which is assumed to be the only objective function. Therefore, SporeME2 was defined based on only the growth rate of the forespore (forespore formation rate). This simplification was critical in solving SporeME2, as it reduces the two-dimensional non-linear programming problem (with two variable growth rates) to a one-dimensional non-linear programming problem, which can be solved using the bisection method described in the ME-model solver package, solveME.”

The authors have clarified that, to simplify the modeling approach, they only account for the growth rate of the foreshore as the community objective function. However, there is also a biomass reaction representing the growth of the mother cell. Considering that this is not part of the objective function, the authors need to explicitly clarify in the text how they are constraining the growth rate of the mother cell (presumably to 0). This clarification is important to ensure the model's assumptions and structure are fully transparent.

(Remarks on code availability)

The installation steps and the content of the repository is much more clear. I appreciate the authors effort to include a docker container. I have installed the docker container and I have run several Jupyter notebooks and scripts. Everything I tried seemed to run nicely.

REVIEWER COMMENTS

Reviewer #1 (Remarks to the Author):

Tibocha-Bonilla and colleagues present an interesting manuscript addressing the metabolic differentiation of *Bacillus subtilis* during sporulation through a multidisciplinary approach. First, a global ME model of the mother cell (MC) and the forespore (FS) is constructed. Then, the model is used to assess the role of key metabolic hubs in cellular differentiation, including nucleotide, energy, and amino acid provision. Some model predictions are elegantly validated using STRP and phase microscopy. Finally, the model is used to predict metabolic reprogramming in the FS through in silico gene essentiality analysis.

The approach has novelty in the field, the manuscript is valuable and includes a substantial workload; however, some sections require further contextualization to be fully exploited and to enhance the current scope of the study. Additionally, greater accessibility to the constructed model is necessary for the research community.

1. While the nucleotides and energy provision is consistent between model and experimental data, the provision of amino acid to FS requires additional clarification, validation, and discussion.

Model predictions suggest minimal intervention of the MC in supplying amino acids to the FS. According supplementary Data 2, the percentage of total production of the most of amino acids is in the FS is 100%. However, these predictions are significantly at odds with the experimental data, which show notable protein dilution for key enzymes in amino acid biosynthetic pathways, including MetE, ArgD, and ArgB. Furthermore, the substantial dilution observed for IlvE indicates that de novo synthesis of branched-chain amino acids is also inhibited in the FS. Which agree with authors conclusions in discussion.

Thank you for catching this. We labeled this column of Supplementary Data 2 in an unclear manner and had intended to show the % of the contribution of each reaction to the FS's requirement of each amino acid, but the header was inaccurate. We have clarified this and also more clearly state in the main text that the model predictions are consistent with previously published data, capturing the inactivity of 25 out of 26 amino acid biosynthetic proteins in the forespore. The manuscript now reads as follows:

“SporeME2 predicts that the mother cell performs amino acid biosynthesis and feeds them to the forespore, although in few cases the transported metabolite is a direct precursor of the final amino acid. [...]”

and

“[...] MS suggests that 26 amino acid biosynthetic proteins are depleted in the forespore. This is largely consistent with our simulations, as 25 out of these 26 were predicted inactive in the forespore, with the only exception of IlvC, which serves a role in the interconversion of threonine coming from the mother cell to isoleucine (see translation flux predictions in Supplementary Data 2).”

Taken together, the simulation results, validation experiments, and further discussions do not fully explain how amino acid provision is managed in the FS, and how the model captures this aspect, if it does. In any case, the major claim by the authors regarding the energetics as model-driven force for amino acid provision is far from being demonstrated in the manuscript.

Could please the authors clarify this point.

Thank you, we have clarified this point by detailing the way that our results describe the provision of amino acids to the forespore (see also answer to the previous reviewer’s comment.

2 Regarding gene essentiality.

-This is a very nice analysis, but it only provides a descriptive overview of model predictions. It would be interesting to contextualize this model-guided analysis within the experimental data. Overall, this analysis introduces some uncertainty in the manuscript, as it does not fully align with previous findings. For instance, nucleotide, sugar phosphate, and amino acid transport appear to be non-essential for the FS. Is this because these three important metabolic hubs are highly robust by design to ensure sporulation even under perturbations? Or is this because the FC MEmodel lacks additional constraints in terms of proteins dilution. Could the author provide any insight on this?

We have now discussed the connection between these findings and the metabolic differentiation that we described in a previous section. Furthermore, we cite two proteomics studies that support our findings that biomass precursor (amino acid and nucleotides) synthesis is dramatically reduced in the FS. As a result, our simulations show that these pathways are inactive, and our essentiality analysis shows that they are non-essential. That said, a deeper comparison with GFP and STRP experiments, though interesting for contextualization, would be currently infeasible due to the time intensity of the experiments. The manuscript reads now:

“The ME2-model allowed us to predict protein activity and essentiality during sporulation, identifying metabolic priorities and key nodes in the metabolic network. We found that the mother cell focuses on energy and amino acid production, while the forespore prioritizes biosynthesis of structural components, likely due to its ongoing transition to dormancy. Furthermore, our essentiality analysis showed that these biomass precursor biosynthetic pathways are not only inactive, but also essential only in the MC, which further supports its nurturing role. Previous proteomics studies showed a significant reduction in amino acid and nucleotide biosynthetic proteins in the spore, which is consistent with our simulations. However, some proteins involved in amino acid biosynthesis were identified in the spore in these studies, which could be explained by them being required for spore germination rather than forespore formation.”

-Would it be possible to perform a synthetic lethality analysis? This approach would help identify potential vulnerabilities in the FS that are not apparent under single perturbation conditions.

This is a great suggestion, as it would allow us to identify metabolic enzymes or pathways that can compensate for the absence of dispensable enzymes or pathways, and in theory one could use both computational and genetic approaches to ensure that the results are consistent. However, this would be labor intensive, as a thorough computational approach would require testing all 1,978 dispensable enzymes against themselves, producing 3,910,506 combinations, which calculates to more than 70 years of computation time (~10 minutes per simulation). On the other hand, the genetic approach would require genetic synergy screens to be performed in 1,978 strains, with significant effort required for each strain. This would clearly be a valuable approach to deploy in the future, as we further refine the model and solve it more efficiently, but it is outside the scope of the present manuscript. We did try this for the proteins involved in ATP synthesis, but as is often the case, strain growth or sporulation defects arose in double knockout or STRP strains that prevented us from using these strains in our experiments.

-Additionally, can the authors contextualize the protein essentiality data with the previous sections of the manuscript and with the available experimental data?

Thank you for identifying a lack of connectivity between these two sections of our results. We have contextualized now these two results in our Discussion section. The manuscript now reads:

“The ME2-model allowed us to predict protein activity and essentiality during sporulation, identifying metabolic priorities and key nodes in the metabolic network. We found that the mother cell focuses on energy and amino acid production, while the forespore prioritizes biosynthesis of structural components, likely due to its ongoing transition to dormancy. Furthermore, our essentiality analysis showed that these biomass precursor biosynthetic pathways are not only inactive, but also essential only in the MC, which further supports its nurturing role. Previous proteomics studies showed a significant reduction in amino acid and

nucleotide biosynthetic proteins in the spore, which is consistent with our simulations. However, some proteins involved in amino acid biosynthesis were identified in the spore in these studies, which could be explained by them being required for spore germination rather than forespore formation.”

3. On the other hands, the author predicted new protein depletions that have not yet been validated, which could significantly impact FS metabolism. Validating such predictions would be interesting and could provide additional insights into amino acid metabolism.

We here chose to highlight the ability of metabolic modeling to predict which redundant metabolic pathway is deployed in certain situations, rather than to test all possible predictions of the model. We here focused on how the forespore produces ATP, which can be produced by an array of oxidative phosphorylation and substrate level reactions, and where our candidate enzyme approach had not succeeded. This is clearly an important path for refining the model, but verifying the 398 proteins required in either the forespore or the mother cell is outside the scope of this model. However, to address this question, we have added a separate section highlighting the predictions we make that are consistent with published data from our lab and others (Swarge et al.¹ and Huang et al.²), and note that this will be critical for future research, highlighting a few pathways we feel are especially important for verification (still need to do this).

4. Related to the code used to generate the model, I encountered difficulties running it according to the provided instructions. To ensure reproducibility of the results, it would be needed to clarify and complete these instructions or resolve the existing issues. For example, I found local folder references, likely from the authors' computers, and the software requires the COBRAME package to be pre-installed, which was not specified.

Improving the accessibility and usability of the code is crucial for the broader research community to validate and build upon this work. Please provide a more user friendly and accessible version of the code.

*We would like to thank the reviewer for pointing out the lack of user-friendly code for better reproducibility and distribution of our computational results. Considering the difficulty in installing the correct package versions and compiling the solver, we now provide two documented ways of installing and running our notebooks, which are now described on the GitHub repository front page (<https://github.com/jdtibochab/sporeme>). The first way is a local installation using a setup script, which requires extensive manual steps. Still, we have kept it, considering it is the standard method reported by other ME-model projects, such as ECOLIME (<https://github.com/sbrg/ecolime>) and BACILLUSme (<https://github.com/jdtibochab/bacillusme>). The second way uses Docker, a user-friendly interface that manages software images and containers and is the go-to choice for computational method reproducibility. We have provided the file “Dockerfile-Python3.7”, which is sufficient to reproduce an environment capable of running and reproducing all of our computational results. We have also outlined the steps for creating a Docker image using the provided Dockerfile and running a container for this purpose. Note that running the container automatically creates a Jupyter Notebook session that can be accessed through **localhost:10000** in any internet browser. The notebook can then be used to run and reproduce all results and figures. Furthermore, we have included a detailed explanation*

of the layout of the repository to point to the specific files and notebooks that contain the key findings of this manuscript. Finally, we have resolved the references to relative paths to avoid errors when running on different computers.

5. Furthermore, I couldn't find the final model, including the mother cell and the forespore, in the provided files. They might be among the large number of files in the GitHub repository. I suggest creating a direct link in the README or a specific folder for the final model to facilitate its identification. This is particularly important because the model represents one of the main advances of this publication, and having easy access to it would greatly benefit the readers and researchers who wish to utilize and build upon your work.

We thank the reviewer for pointing out the difficulty in understanding the layout of this repository. To resolve this, we have included a detailed explanation of the layout of the repository, pointing to the specific files and notebooks that contain this manuscript's key findings.

Reviewer #1 (Remarks on code availability):

Related to the code used to generate the model, I encountered difficulties running it according to the provided instructions. To ensure reproducibility of the results, it would be needed to clarify and complete these instructions or resolve the existing issues. For example, I found local folder references, likely from the authors' computers, and the software requires the COBRAME package to be pre-installed, which was not specified.

Improving the accessibility and usability of the code is crucial for the broader research community to validate and build upon this work. Please provide a more user friendly and accessible version of the code.

Furthermore, I couldn't find the final model, including the mother cell and the forespore, in the provided files. They might be among the large number of files in the GitHub repository. I suggest creating a direct link in the README or a specific folder for the final model to facilitate its identification. This is particularly important because the model represents one of the main advances of this publication, and having easy access to it would greatly benefit the readers and researchers who wish to utilize and build upon your work.

We want to thank the reviewer for pointing out a key obstacle in our repository that impedes a quick and user-friendly reproduction of our computational results. We have now resolved this as described in the reviewer's previous comment.

Reviewer #2 (Remarks to the Author):

Reviewer #3 (Remarks to the Author):

The authors sought to address the long-standing question of how the forespore produces the energy needed to complete sporulation.

Using the model prediction approach, the authors identified 1,707 proteins, 11,389 reactions, and 7,440 metabolites both in both the mother cell and forespore compartments. They proposed that the sporulation-specific SpoIIQ-SpoIIIA complex (Q-A) transports metabolites from the mother cell to the forespore.

Using several of the C-terminal tagged GFP enzymes (~16), the authors experimentally demonstrated compartment specific localization of the tested proteins.

STRP-mediated depletion of enzymes in the mother cell or forespore was used to assess the essentiality of enzymes involved in the synthesis of NDPs or NTPs in one or both cells. The enzymes analyzed were PyrG (CTP synthase), Gmk (guanylate kinase), and Cmk (cytidylate kinase).

Strains that degrade PyrG, Gmk, or Cmk in the forespore are able to produce spores but do not outgrow, suggesting that these enzymes are essential for vegetative growth.

The results indicated that degradation of PyrG, Gmk, or Cmk is dispensable in the forespore, suggesting that NDPs or NTPs synthesized in the mother cell are transported to the forespore through the Q-A channel or an unidentified channel.

The results suggest that the enzymes required to produce GTP, CTP, and UTP are only required only in the mother cell for spore maturation, consistent with a model in which NTPs are transported to the forespore via the Q-A channel or an unidentified channel.

They proposed that if the Q-A channel is inactivated after engulfment, energy (ATP) could be produced in the forespore by glycolytic enzymes. Thus, the authors concluded that glycolytic enzymes transfer ATP to the forespore.

The authors interpreted their results by suggesting that if the Q-A channel is inactivated after engulfment, ATP could be produced in the forespore using glycolytic enzymes, including Pyk.

Specific questions and concerns.

The synthesis of housekeeping factors, including core subunits of RNA polymerase and ribosome subunits, and essential metabolites, such as NTPs and amino acids, is mainly controlled by the housekeeping SigmaA RNA polymerase, which is generally known to be inactivated during sporulation when the sporulation-specific Sigma factors (SigmaE, K, F and G) become active (PMID: 10200951, 7708009, 4198276, Subtiwiki website). NTPs, including ATP and GTP, amino acids, RNA polymerase core enzyme, and ribosome subunits are essential for the synthesis of mRNAs and proteins in the forespore. Therefore, the available data suggest that most of these essential housekeeping components are synthesized prior to polar septum formation and that initial concentrations of these components distributed in the sporangium are sufficient to complete sporulation. Thus, it is not clear whether the authors' model of "the forespore producing the energy needed to complete sporulation" is correct. Overall, I feel that the model proposed here is not necessarily supported by the published and available data. The authors need to address these uncertainties.

Thank you for pointing out that our text did not clearly specify that even though these proteins are produced prior to polar septation, that some enzymes are actively depleted from the forespore (Riley et al.³, Riley et al., in preparation), and thus, their levels in the forespore may not suffice to support sporulation. We have emphasized this in the introduction to make this more clear and hope that we have addressed this concern.

While SspB is expressed specifically in the mother cell and the forespore, the subunits of ClpXP are expressed under the control of SigmaA RNA polymerase. Again, SigmaA is inactivated in the forespore compartment. Therefore, the subunits of ClpXP appear to be synthesized before polar septation and to be distributed to the forespore compartment. As a result, the protease activity remains in the forespore compartment after polar septation. Based on these observations and facts, the housekeeping factors are not necessarily synthesized in the forespore compartment. The authors need to investigate this paradox.

Again, thank you for pointing out this area where our text was unclear. ClpXP is indeed synthesized before septation from Sigma A, and remains in both cells after division. However, the E. coli SspB adapter protein that is required for ClpXP degradation of proteins tagged with the E. coli ssrA peptide is not present in Bacillus subtilis, so we can express SspB from cell-specific promoters, allowing ssrA-tagged enzymes that are present in both cells to be specifically degraded in just one cell by ClpXP. This allows us to alter the metabolic network of the forespore and mother cell independently, without affecting cell growth or early steps in sporulation. We clarified this section of the manuscript, and hope this has answered the reviewer's question.

The reviewer is correct that only a few housekeeping enzymes are synthesized from forespore specific promoters (these were integrated into the metabolic model following annotation from BioCyc and SubtiWiki, as per Tibocho-Bonilla et al.⁴), and that sigma A is unlikely to be active during sporulation. Indeed, we have shown that sigma A is dispensable in the mother cell and forespore for spore assembly although it is required in the spore for germination⁵. This suggests that most housekeeping enzymes required in the forespore are likely synthesized before cell

division. However, we have shown that some of these housekeeping proteins are specifically depleted from the forespore, which motivated the present study as a means to better understand this example of coupled metabolism between two differentiated cells.

Glycolytic genes, including pfkA, fbaA, tpiA, gapA, pgk, pgm, eno, and pyk, are not essential (PMID: 23420519). These results suggest that, under certain conditions, ATP synthesis may be achieved by other pathways, not primarily by glycolytic enzymes. This possibility needs to be investigated by the authors.

The reviewer is correct that ATP synthesis can be accomplished by many different pathways, this was exactly our motivation to address these open questions through a ME-modeling approach, which accounts for all metabolic costs and evaluates every pathway possible. The lack of essentiality and redundancy of ATP synthesis thus necessitates a model. In our model, we indeed found that individual knockouts of any of the predicted players in the ATP production pathway would readily be compensated by other alternative pathways, meaning that even though one pathway can be predicted as optimal, other alternative ones can also come into play. We have now updated our Figure 3a to further represent this.

The gapA gene in the hexacistronic cggR gapA pgk tpi pgm eno operon (the gapA operon) is repressed by CggR, the product of the first gene in the absence of glucose or other sugars and is expressed only in the presence of glucose (PMID: 11489127, 10799476). These results do not support the data presented by the authors in this study (Fig. 3). The authors need to comment on this issue.

Thank you for pointing this out. While transcriptional regulation is not explicitly included in metabolic models or ME-models, the model does recapitulate that there must be sugar metabolites upstream of glycolysis in order to run gapA (glycolytic direction). Indeed, our ME model predicts that fructose 6-P is the predominant metabolite transported to the forespore, but that glucose 6-P can also be transported, and likely must be present to allow expression of gapA. We have modified the figure to make this more clear.

"Amino acid supply to the forespore is driven by energetics"

"Protein essentiality predictions reveal metabolic reprogramming at the proteome level"

"Prediction of new possible proteome-scale depletions".

These topics have only been predicted by modeling. Authors must provide experimental data to support these predictions.

We thank the reviewer for pointing out a lack of clarification in our results. We now cite two proteomics studies that support our findings that biomass precursor (amino acid and nucleotides) synthesis is dramatically reduced in the FS. As a result, our simulations show that these pathways are inactive, and our essentiality analysis shows that they are non-essential. That said, a deeper comparison with GFP and STRP experiments, though interesting for

contextualization, would be currently infeasible due to the time intensity of the experiments. The manuscript reads now:

“The ME2-model allowed us to predict protein activity and essentiality during sporulation, identifying metabolic priorities and key nodes in the metabolic network. We found that the mother cell focuses on energy and amino acid production, while the forespore prioritizes biosynthesis of structural components, likely due to its ongoing transition to dormancy. Furthermore, our essentiality analysis showed that these biomass precursor biosynthetic pathways are not only inactive, but also essential only in the MC, which further supports its nurturing role. Previous proteomics studies showed a significant reduction in amino acid and nucleotide biosynthetic proteins in the spore, which is consistent with our simulations. However, some proteins involved in amino acid biosynthesis were identified in the spore in these studies, which could be explained by them being required for spore germination rather than forespore formation.”

Intracellular metabolite concentrations have been determined experimentally in *E. coli* (PMID: 19561621). Can you estimate the concentrations of key metabolites and the energy requirements for cell division and spore development? The authors need to investigate these points to estimate the concentrations of the metabolites in *B. subtilis*.

Our study did not perform modeling on dividing growing cells, as it was focused on the coupled metabolism of sporulating cells after synthesis of the sporulation septum. We agree that it would be useful to compare the metabolome of the forespore and the mother cell, but unfortunately this more complicated than in growing cells, because in a sporulating culture only 50% of cells are sporulating, and among the sporulating cells, ~80-90% of the cell volume is the mother cell with just 10-20% from the forespore. Thus the metabolome would represent a mix of approximately 50% non sporulating starving cells, 40% mother cell and 10% forespore. We do not currently have a means to physically separate the forespore and the mother cell or to perform single cell metabolomics, but look forward to further development of these techniques.

References for PyrG CTP synthase (essential, SigA controlled), Gmk guanylate kinase (GTP biosynthesis, essential, SigF controlled), and Cmk cytidylate kinase (cytidylate kinase, essential) need to be added for results and discussion, such as expression before and after polar septation.

Great suggestion. We have added this reference⁶.

The authors describe: “Together, these results suggest that the Q-A channel, or an unidentified channel, transports NMPs, NDPs, or NTPs to the forespore during spore formation”. Is it possible that there is an unidentified channel that transports NTPs? An unidentified channel would be essential for sporulation if it existed, but such "sporulation" genes have never been identified. The authors need to clarify this.

We would like to thank the reviewer for pointing out a lack of clarification in the QA, its role and essentiality. The unidentified channel may not be essential for sporulation, because of the many backup pathways to synthesize ATP, including the glycolytic shuttle that we propose is activated

after the Q-A channel is inactivated. In addition, new genes required for efficient sporulation continue to be identified, for example, Chen et al 2022 DOI: 10.1128/mbio.01732-22.

The channel(s) would be a passive transport system. Please discuss this to exclude the possibility of active transport or any other types of transport.

We agree that it is likely that the Q-A channel is a passive channel, but this has not been experimentally tested. There is an ATPase in the complex that likely participates either in complex assembly or translocation. It is not clear to us that metabolites would need to be transported by a passive transport system, because cells are able to synthesize ample quantities of ATP during sporulation and support energy-driven transport systems, as demonstrated by the SpoIIIE DNA translocase, which translocates 3 megabases of DNA across the sporulation septum into the forespore in ~10 minutes while hydrolyzing one ATP for every 2 bp translocated.

Experimental data on genes controlled by compartment specific sigma factors have been published (PMID: 10869437, 12169614, 16497325, 15383836, 12662922). The authors must mention how these published data support the authors' model prediction, such as the expression of glycolytic enzymes in the forespore compartment during sporulation.

Thank you, we built the model based on this data and have now more clearly specified this in the text. The manuscript now reads:

*“Our ME2-model of *B. subtilis* sporulation, SporeME2, was based on a recently published ME-model of *B. subtilis*, iJT964-ME. As a result, our ME2-model follows all the assumptions and formulations of COBRAME-based ME-models¹⁴. Furthermore, it includes metabolism and gene expression functions of *B. subtilis*, including transcription unit sigma factor specificity, as supported by BioCyc, UniProt, and SubtiWiki”*

In Fig. 3C, how were these 16 proteins selected from many others for the GFP-tagged experiments. The imaging data would be helpful to verify the data in Fig. 3C.

We thank the reviewer for catching the missing information in the selection of the 16 proteins. This data is included in supplemental figure 2, and we now more clearly refer to this supplemental figure in the main text and figure legend.

Reviewer #4 (Remarks to the Author):

Dear authors,

I have read with great interest your manuscript which I think nicely highlights the interdisciplinary nature of the work where modelling, spatiotemporal-regulated proteolysis (STRP) and GFP-tagging meet once more. The 2021 Science Advances paper was in that sense more ground-breaking as it is a fully new view on the cellular differentiation at the molecular metabolic level. The current paper is in a way a specialized further detailing of the approach pioneered in the Science Advances paper. In fact, the current focus is on the Q-A channel and its role in energy metabolism of the fore spore as well as on shuttles and proteins putatively involved in

these. While the enzymes involved in gluconeogenesis and glycolysis partition as can be logically explained from a mother cell forespore nurturing perspective, expected the presumed membrane transporter proteins have not (yet) been unequivocally identified. Can the authors comment on/ discuss this.

Additionally, the amino acid and protein biosynthesis pathways are modelled and analyzed. The logic of protein biosynthesis in the fore spore and amino acid production supplying the fore spore is compelling and illustrated nicely with the experiments done.

The overall focus on metabolism as a key element in understanding cellular differentiation is laudable and unique. It is remarkable though that the authors do not put their findings in perspective of the proteomics and metabolomics data on spore and vegetative cell composition in *Bacillus* as published in *mSphere* in 2020 and this year in *J. Proteome Res.* (Huang et al. 2024). This omission should be adjusted.

Thank you for pointing this out, discussing our results in light of these studies complemented our analysis. We have contextualized our analysis with the data presented in the two highlighted papers and have updated this contextualization throughout the manuscript accordingly.

Attached to this review the annotated manuscript contains still some specific comments to address.

Reviewer 4's comments from the PDF:

- **But where does the lactate come from? Environment (other cells that are lysed late into sporulation) or mother cell that 'simply' reverses previous glycolytic flux from the fore-spore? I take note of the thesis that it is the latter primarily but is there experimental evidence (radioactive tracing?).**

Thank you for pointing out the lack of explanation when we introduce the lactate flux in the manuscript. More specifically, we mentioned it when describing the predicted ATP shuttle, and explained it further in the next paragraph that the source of lactate is reverse glycolytic flux. We have now removed the first mention so that the concept and its explanation are introduced at the same time. The manuscript reads as follows:

“However, our simulations show that the mechanisms shown in Fig. 3a are beneficial due to the additional production of ATP through P_{gk} and SucCD. These additional steps produce an excess of NADH in the forespore by GapA (44%) and PdhD (55%), which is balanced by an additional step of converting pyruvate to lactate via LctE.”

- “Therefore, our results suggest that if the Q-A channel is inactivated after engulfment, then energy could be produced in the forespore via a mechanism using glycolytic enzymes.” - **Modelling and protein/ enzyme correlative analyses.**
- “GFP intensity levels support this hypothesis, as core biosynthetic enzymes in amino acid metabolism, such as MetE, ArgD, and ArgB, were depleted from the forespore (Fig. 3c).” -

Comparison with the data of Swarge et al. 2020 mSphere and Huang et al. 2024 J Prot Res is needed.

We would like to thank the reviewer again for introducing the two relevant publications by Swarge et al. and Huang et al. As mentioned previously, we have now included the contextualization of these two studies in our results. In this specific case, the manuscript now reads:

“GFP intensity levels support this hypothesis, as core biosynthetic enzymes in amino acid metabolism, such as MetE, ArgD, and ArgB, were depleted from the forespore (Fig. 3c). Notably, previous proteomics reports have also reported the inactivation of the amino acid biosynthetic pathways in the forespore.”

- “In contrast, 42 proteins predicted to be essential in the forespore were required mostly for the synthesis of structural components (Fig. 4a), were involved in cell wall and membrane biosynthetic pathways (Fig. 4b).” - **Sentence does not run well.**

Thank you we corrected this.

- “These results suggest that no single depletion of the confirmed ones is impactful enough to differentiate the metabolism of the forespore, but rather, the combination of all depletions is critical for differentiation.” - **This conclusion is not 100% clear to me please elaborate!**

Thank your for pointing out the lack of clarity in this statement. We have now expanded the detail as follows:

“These results suggest that, out of the 13 confirmed depletions, no single depletion is impactful enough to inactivate or block entire pathways that fully differentiate the metabolism of the forespore. Rather, it is the combination of several depletions that metabolically differentiates the forepore.”

- **OK so the thought is that the lactate originates from the forespore; how long into sporulation does the mother cell keep on nurturing the forespore? Compared to the Cori cycle in metabolism the purpose seems a different one i.e. survival structure synthesis though it could be argued that muscle activity is ultimately also there for survival!**

We do not know how long the mother cell continues to nurture the forespore and while it is not the focus of the paper, it is an interesting question and we can speculate. Our current thinking is that most of the nurturing comes from the mother cell passing metabolites and other molecules to the forespore through the SpoIIQ-SpoIIIA channel. If that is the case, then we know through previous studies (Chiba et al 2007) that SpoIIIA is cleaved before engulfment is complete, perhaps closing the channel in the process. In this event, it is likely nurturing stops here, at engulfment completion.

Interesting thought about the Cori cycle. We see the Cori cycle as similar to the energy recycling shown here in that they both involve one cell/organ taking the metabolic burden for another more important to the current situation cell/organ.

- “compared the enzyme composition of Bacillus spores to vegetative cells and found many enzymes missing or in low levels in the forespore. Since” - **The omission of the state of the art work modern work on proteins and metabolites in spores and cells of Bacillus should be corrected. See Huang et al. ACS's J Proteome Research 2024.**

We thank the reviewer for bringing up these two studies. We have now contextualized our conclusions in light of their findings.

- “Our study also identified key enzymes whose metabolic flux is shut down by depletions, and many of these predictions still remain to be confirmed. These enzymes could be targets for the forespore-specific proteolytic machinery since their degradation would shut down flux through other pathways, allowing the “unemployed” enzymes to be degraded by the Clp proteases of B. subtilis, which have been shown starvation. to degrade non-functional “unemployed” enzymes during” - Open functional questions!

The reviewer is correct in the statement that not all functional questions have been answered by our work and that there are still questions to be addressed in future studies. We strongly believe that the approach presented here provides a new method to achieve a broader understanding of spore formation and cell differentiation in the future.

Reviewer #5 (Remarks to the Author):

Deciphering metabolic differentiation during Bacillus subtilis sporulation

Tibocha-Bonilla, Lyda et al. follow an in silico and in vivo approach to study sporulation in Bacillus subtilis. Sporulation involves an asymmetric cell division producing two distinct cells - a large mother cell and a small forespore; each tightly regulated by gene expression. A recent study hypothesized that the forespore is highly dependent on the mother cell, since the forespore needs building blocks produced by the mother cell for protein synthesis required to complete spore assembly. Motivated by the latter hypothesis, the authors present a novel approach to study cross-feeding interactions between mother cell and forespore. For that, they extrapolate the principles of community modeling framework used with metabolic models (M-models/GEMs) to model cell differentiation by using a community metabolic and gene expression (ME-) model of the two cell types. They construct a community ME-model (SporeME2) made by two ME-models, one representing the mother cell and one representing the forespore, allowing metabolite cross-feeding between the two cells.

Model predictions were then tested by in vivo analyses making the study more robust. Predicted metabolic interactions between the two cells were used to guide spatiotemporal regulated proteolysis (SPTR) and C-terminal GFP-tagging experiments and further fluorescence microscopy. Following this approach the authors gain new insights into the mechanisms of

spore assembly and cell differentiation in *B. subtilis*. Furthermore, the presented findings can help expanding the understanding of the same mechanisms in other organisms.

While the study is innovative and present valuable insights and contributions to the field, the authors need to clearly explain, develop and justify the methodology used to facilitate the understanding, validation and reproducibility of their modeling results. Below there is a list of comments that the authors need to address to achieve that.

Major comments

1.Methods. The authors indicate that they use a similar approach to community modeling. In a community model (M-models), each species has their own biomass reaction, and each species is represented as a single compartment. One can then create a community biomass reaction by adding the weighted contribution of the biomass of each species (using their respective metabolite), and set this reaction (if applicable) as the objective function. Many studies and tools have shown that integrating the relative abundance/biomass species ratio within the community modeling framework is key to correctly applying this approach (<https://journals.plos.org/ploscompbiol/article?id=10.1371/journal.pcbi.1005539>, <https://journals.asm.org/doi/10.1128/msystems.00606-19>, <https://journals.plos.org/ploscompbiol/article?id=10.1371/journal.pcbi.1011363>), as it might also lead to different flux distributions.

While the framework used in e.g SteadyCom is not compatible to ME-models, due to the non-linearity issue, one can constrain the growth rates and scale the exchange fluxes by the strain/cell relative's abundance (<https://www.sciencedirect.com/science/article/pii/S2001037020304256>). In fact, this methodology has been implemented already in a community model using ME-models (<https://journals.plos.org/ploscompbiol/article?id=10.1371/journal.pcbi.1006213>).

Considering the authors are using a community model to study the interactions between the two cells, and the two cells differentiate in volume, and thus, in biomass, the authors need to clarify how they account (if they do) for the relative abundance in their modeling framework. If the authors do not account for the relative abundance, they need to justify their methodology used to validate their approach.

We thank the reviewer for bringing up relevant community modeling strategies that were not properly contextualized in the manuscript. As noted by the reviewer, relative abundances have been used in ME-models to constrain the growth rates, thus linearizing the problem and allowing for a more standard community modeling strategy. Relative abundance-based community modeling is very powerful for microbiome studies. However, estimating the relative abundances (in terms of mass) of mother cells and forespores is difficult. While it would be an

interesting approach, we currently do not require linearizing our model by estimating growth rates from the relative abundances since the “community objective” is the maximization of the growth rate of the forespore. This greatly simplifies the objective function and allows the use of the traditional one-dimensional bisection method described in solveME. We now include a new paragraph in methods providing this information more clearly. The paragraph reads:

“Regarding the second question, our system only optimizes for the formation rate of the forespore, which is assumed to be the only objective function. Therefore, SporeME2 was defined based on only the growth rate of the forespore (forespore formation rate). This simplification was critical in solving SporeME2, as it reduces the two-dimensional non-linear programming problem (with two variable growth rates) to a one-dimensional non-linear programming problem, which can be solved using the bisection method described in the ME-model solver package, solveME.”

Furthermore, we now contextualize our work and mention the previous report on community ME-models. The paragraph in the Introduction reads:

“While M-models have been used to study the metabolic interactions of two or more organisms¹⁵, multi-cell ME-models have remained largely unexplored, with only one study generating a multi-strain E. coli community ME-model to design syntrophic co-cultures.”

2. Following the previous comment, the authors seem to tackle that matter by only considering growth of the forespore in SporeME2 model. The code shows a spore biomass reaction and a section that mentions ‘Mother is not growing’ (BuildCommunityModel.ipynb). Considering the mother cell produces the main building blocks (biomass precursors) for the forespore, which is a sign of cell growth, it is not clear how the authors represent growth (if represented) on the mother cell. Does the mother cell have a separate biomass reaction? How is it defined and constrained in the model? The authors need to clarify this point in the text and justify their modeling framework.

We thank the reviewer for catching the lack of description of the objective function and the solution strategy in the manuscript. Precisely, both cells present growth rates, as the reviewer correctly inferred, because they produce expression machinery, which must be diluted to biomass in order to keep the steady state. However, the key simplification that allowed us to solve this model is that only the forespore’s growth rate is optimized, giving an optimization problem that is still non-linear but only one-dimensional. In this way, we were able to use the bisection method described previously in the solver package, solveME. We have now included a paragraph in the methods providing these details, and have also updated the notebook’s misleading details. The paragraph reads:

“Regarding the second question, our model only optimizes for the formation rate of the forespore, which is assumed to be the only objective function. Therefore, SporeME2 was defined based on only the growth rate of the forespore (forespore formation rate). This simplification was critical in solving SporeME2, as it reduces the two-dimensional non-linear programming problem (with two variable growth rates) to a one-dimensional non-linear

programming problem, which can be solved using the bisection method described in the ME-model solver package, solveME.”

- Change wording in text

3. Properties of the SporeME2 model ...

The authors mention: ‘Supplemental Data 1 contains the flux predictions’. However, it is not clear what are model predictions indicating here, and how are the fluxes obtained in the model, under which conditions, what are the import fluxes or how they are constrained (in methods). The authors need to clarify all this information in the text to facilitate the understanding and reproducibility of their results.

We thank the reviewer for identifying the lack of description of the base simulation conditions, including medium composition, that leads to the presented flux distribution results. We have included a paragraph in the methods section outlining these details. The paragraph reads:

*“The simulation conditions were set to resemble the experimental sporulation conditions (see Culture Conditions). Thus, we simulated a minimal medium composed of salts and supplemented with glutamate, with a lower bound of -2 mmol/gDW/h. This bound was set to a comparable value to the original glucose uptake bound in the M-model of *B. subtilis*, -1.7 mmol/gDW/h in *iYO844*, which allows for a typical growth rate of 0.1 h^{-1} . We allowed the model to uptake several metal ions provided in the medium. The full definition of the medium used is provided in Supplementary Data 1.”*

4. The installation of sporeme is not sufficiently clear in the github repository. The authors need to add the cobrame installation within the installation steps. There is a lack of information on the different versions required to run sporeme. This creates compatibility issues (e.g python, numpy, optlang..) when first running python setup.py develop –user, suggesting this is not the most appropriate method.

This could be solved by creating a virtual environment in conda with the specific versions, or using the docker image in COBRAME.

However, the main issue is with cobra. The authors recommend to use cobra 0.5.11, but this version is not available when using pip install cobra=0.5.11, and neither in bioconda. Therefore, it is crucial that the authors solve this issue upon validation of their results.

We thank the reviewer for taking the time to identify key problems in the installation, the easy reproducibility of the computational environment, and the manuscript’s computational results. Considering the difficulty in installing the correct package versions and compiling the solver, we now provide two documented ways of installing and running our notebooks, which are now described on the GitHub repository front page (<https://github.com/jdtibochab/sporeme>). The first way is a local installation using a setup script, which requires extensive manual steps. Still, we have kept it, considering it is the standard method reported by other ME-model projects, such as ECOLIme (<https://github.com/sbrq/ecolime>) and BACILLUSme (<https://github.com/jdtibochab/bacillusme>). As the reviewer pointed out, it also requires manual

installation of COBRAPy 0.5.11, as it was unexpectedly taken down from PyPI and, thus, cannot be installed using pip (only through the COBRAPy's GitHub release history). The second way uses Docker, a user-friendly interface that manages software images and containers. We thank the reviewer for suggesting it to resolve the issues of distribution and reproducibility. We have provided the file "Dockerfile-Python3.7", which is sufficient to reproduce an environment capable of running and reproducing all of our computational results. We have also outlined the steps for creating a Docker image using the provided Dockerfile and running a container for this purpose. Note that running the container automatically creates a Jupyter Notebook session that can be accessed through **localhost:10000** in any internet browser. The notebook can then be used to run and reproduce all results and figures. Furthermore, we have included a detailed explanation of the layout of the repository to point to the specific files and notebooks that contain the key findings of this manuscript. Finally, we have resolved the references to relative paths to avoid errors when running on different computers. We would like to point out that we have included the compiled quad-precision solver files as part of our effort to make the results reproducible using Docker, which can also be used for manual installations (<https://github.com/jdtibochab/sporeme/tree/main/solver>).

Minor comment

1. Results. Properties of the SporeME2 model ...

The authors seem to build the SporeME2 by aggregating two ME-models. The mother cell model seems to be the iJT964-ME plus transporters, and the forespore model is adapted from the original iJT964-ME by correcting for protein depletions and allowing transport through A-Q complex. This is clear in the methods and Figure 1.c, but the results could benefit of a brief explanation too.

We thank the reviewer for detailing a lack of explanation of the transporter configuration in SporeME2 in the results. We have expanded these details in the Results section, as follows:

*"We adapted the existing *B. subtilis* ME-model, iJT964-ME¹⁶, to build a ME2-model representing the connected mother cell and forespore (Fig. 1a). Each model contained a metabolic network and gene expression network (Fig. 1b), with the forespore model excluding 13 proteins identified to be depleted in the forespore by mass spectrometry and validated through a GFP localization assay to be depleted in the forespore but present in the mother cell (data are available via ProteomeXchange¹⁷ with identifier PXD051727) (Fig. 1c, Supplementary Fig. 1). Furthermore, we allowed for transport of all metabolic intermediates via the sporulation-specific SpoIIQ-SpoIIIA complex (Q-A)^{10-12,18-20} that has been shown to facilitate transport of calcein⁸, which is larger than most metabolic intermediates. In brief, the mother cell and the forespore ME-models inherited the stoichiometric matrix from iJT964-ME. Furthermore, expression reactions for 13 identified protein depletions were closed in the forespore ME-model, which was then connected with the mother cell by transport reactions through the Q-A complex."*

2. The authors need to clarify in the text how are the transport of intermediates via the Q-A complex represented in the model.

We have now included a more detailed explanation of the transport of metabolites in our model in the Online Methods section, which reads:

*“All transport reactions were implemented from transporters in the ME-model of *B. subtilis*, iJT964-ME¹⁶. We then allowed transport via the sporulation-specific SpoIIQ-SpoIIIA complex (Q-A)^{10,11,18,19}. We kept the transporter stoichiometries from iJT964-ME and replaced the catalyzing complex with the Q-A complex. It is worth noting that only those intermediates transported in iJT964-ME were allowed to be transported by Q-A in SporeME2.”*

3. The intracellular reactions of each species are defined in two cytosolic compartments as mentioned by the authors, and metabolites can be transported from the cytosolic compartment of the mother cell to the extracellular compartment and from the extracellular compartment to the cytosolic compartment of the forespore and vice versa. However, the cytosolic compartments need to be defined with different ids ('_c', '_s'), so can the authors briefly clarify this in the text? Are there more compartments in the model as shown in the code?

We thank the reviewer for identifying the lack of detail on the model's compartmentalization, we have now included the following details in the Online Methods section:

“...We then combined the mother cell and forespore ME-models by creating separate compartments and adding all forespore and mother cell metabolites and reactions to the ME2-model in their respective compartments. This resulted in a total of three compartments in the model: the mother cell's cytosol (c), the forespore's cytosol (s), and the extracellular environment (e)...”

4. Results. The authors need to clarify to the reader how are model predictions indicating that spore assembly can be achieved.

We would like to thank the reviewer for identifying the ambiguity in mentioning spore growth and spore assembly. As a metabolic model, SporeME2 can only predict spore growth as the production of individual biomass components. We used this growth as a proxy for the forespore formation process or spore assembly. To disambiguate, we have included this explanation in the first subsection of the Results, which reads:

“... In this model, forespore growth rate is used as a proxy for the complete process of forespore formation or spore assembly. However, it is worth noting that, as a ME-model, SporeME2 represents forespore formation through the biosynthesis of all individual biomass components and does not account for the physical process of structural assembly.”

5. Introduction: The authors refer to 'Recent studies' but then they only use one reference (8). Then they talk about 'The publication used ..'. The authors can cite other studies or rephrase the sentence.

We thank the reviewer for catching the error in this phrasing, it has now been corrected in the manuscript.

6. Introduction. 'multi-cell ME-models are yet to be explored'. Since there are yet not many studies where they implement community ME-models, it is fair that the authors refer to the previous study where they implement community modeling with ME-models (<https://journals.plos.org/ploscompbiol/article?id=10.1371/journal.pcbi.1006213>).

We thank the reviewer for bringing this closely related study to our attention, we have now included this reference in the manuscript's Introduction. It now reads:

"While M-models have been used to study the metabolic interactions of two or more organisms¹⁵, multi-cell ME-models have remained largely unexplored, with only one study generating a multi-strain E. coli community ME-model to design syntrophic co-cultures."

7. The authors add a github repository with the SporeME2 model and all the scripts and data used in the study. However, one does not realize the existence of this repository till the end of the manuscript. For clarity to the reader and further use of this approach, the authors can refer to the existence of this repository in the main text.

We want to thank the reviewer for pointing out a key obstacle in our repository that impedes a quick and user-friendly reproduction of our computational results. We have now resolved this as described in our response to the reviewer's previous comment.

8. Check if the reference to the Supplementary Fig. 3 is correct in the 'STRP and GFP-tagging interrogate mother cell...' section.

We are grateful to the reviewer for pointing out the error in the Supplementary Information file. We have now updated the SI file accordingly.

9. Discussion. 'Since then, mechanisms derived from those absent or diluted enzymes have been minimally studied (4), in part due to the lack of appropriate tools (7)'.

Correct the format of the two references here.

We thank the reviewer for catching this error in the reference formatting, we have updated the manuscript accordingly.

10. Model reconstruction and assumptions. Correct reference 4 and 9 in this section.

We thank the reviewer for catching this error in the reference formatting, we have updated the manuscript accordingly.

Reviewer #5 (Remarks on code availability):

I ran the code in a linux machine following the installation steps. First I got the error for the cobrame not being installed. So, I read the readme file again where they mention cobrame (before the installation steps). So, the installation of cobrame (or link) and required solvers should be clearly stated in the installation steps. Once I installed cobrame I got an issue running 1.1.BuildCommunityModel.ipynb due to the a compatibility issue with the numpy version. Then I checked, and the version you need is 1.15 that seems to work only until python 3.7, which is not something one realized right away. So I created a conda environment and install a downgraded version of python (3.7) and the compatible version of numpy (1.15).

Then I tried to run the 1.1.BuildCommunityModel.ipynb again, and I had to install tqdm. I then got an issue with optlang. I installed it, and again one needs to check which version is compatible to run with the rest of packages. Then I got this error:

```
/anaconda3/envs/sporeme/lib/python3.7/site-packages/cobra/io/__init__.py:10: UserWarning: cobra.io.sbml requires libsbml
```

```
warn("cobra.io.sbml requires libsbml")
```

```
cobrame/__init__.py:30 UserWarning: COBRAPy version is 0.5.4. We recommend using 0.5.11. Using earlier versions may cause errors
```

So I tried to install cobra version 0.5.11 .

However, there is no such version:

```
pip install cobra==0.5.11
```

```
ERROR: Could not find a version that satisfies the requirement cobra==0.5.11 (from versions: 0.2.0, 0.2.1, 0.3.0b1, 0.3.0b2, 0.3.0b3, 0.3.0b4, 0.3.0, 0.3.1, 0.3.2, 0.4.0a1, 0.4.0a2, 0.4.0a3, 0.4.0a4, 0.4.0b1, 0.4.0b2, 0.4.0b3, 0.4.0b4, 0.4.0b6, 0.4.0b7, 0.4.0, 0.4.1, 0.4.2b1, 0.4.2b2, 0.5.1b1.post14, 0.5.2b3, 0.5.2, 0.5.3.post3, 0.5.4, 0.8.2, 0.9.0, 0.9.1, 0.10.0a1, 0.10.0, 0.10.1, 0.11.0, 0.11.1, 0.11.2, 0.11.3, 0.12.0, 0.12.1, 0.13.0, 0.13.1, 0.13.2, 0.13.3, 0.13.4, 0.14.0, 0.14.1, 0.14.2, 0.15.0, 0.15.1a0, 0.15.1, 0.15.2, 0.15.3, 0.15.4, 0.16.0, 0.17.0, 0.17.1, 0.18.1, 0.19.0, 0.20.0, 0.21.0, 0.22.0, 0.22.1, 0.23.0, 0.24.0, 0.25.0, 0.26.0, 0.26.2, 0.26.3, 0.27.0, 0.28.0, 0.29.0)
```

```
ERROR: No matching distribution found for cobra==0.5.11
```

and neither in bioconda.

```
cobra 0.4.0b6 py27_0 bioconda
```

```
cobra 0.4.0b6 py34_0 bioconda
```

```
cobra 0.4.0b6 py35_0 bioconda
```

cobra 0.4.0 py27_0 bioconda
cobra 0.4.0 py27_1 bioconda
cobra 0.4.0 py34_0 bioconda
cobra 0.4.0 py35_0 bioconda
cobra 0.4.0 py35_1 bioconda
cobra 0.4.0 py36_0 bioconda
cobra 0.4.0 py36_1 bioconda
cobra 0.10.1 py27_0 bioconda
cobra 0.10.1 py35_0 bioconda
cobra 0.10.1 py36_0 bioconda
cobra 0.10.1 py_1 bioconda
cobra 0.15.4 py_0 bioconda

And then I stop here because I cannot run it anymore if the only recommended version to run cobra is not available anymore.

Based on this revision:

```
python setup.py develop --user
```

would not be the most appropriate command as it will install different versions that are not compatible with one another. So, all the dependencies and versions should be clearly stated.

Alternatively, COBRAME link mentions the possibility of working with a docker image, which would help to solve the compatibility issues.

However, I don't know if it would work, as it happens when creating a conda environment.

It is essential that the authors correct these issues in order to validate their method, and thus, their results. Once they do it, they need to extend and clarify the installation steps and all the dependencies required to properly run sporeme.

We want to thank the reviewer for pointing out a key obstacle in our repository that impedes a quick and user-friendly reproduction of our computational results. We would also like to thank the reviewer for suggesting using Docker, which was key in resolving this issue in our proposed solution. Please refer to the details in the previous reviewer's comment.

References

1. Swarge, B. *et al.* Integrative analysis of proteome and transcriptome dynamics during *Bacillus subtilis* spore revival. *mSphere* **5**, 10.1128/msphere.00463-20 (2020).
2. Huang, Y. *et al.* Integrative metabolomics and proteomics allow the global intracellular characterization of *Bacillus subtilis* cells and spores. *J. Proteome Res.* **23**, 596–608 (2024).
3. Riley, E. P., Lopez-Garrido, J., Sugie, J., Liu, R. B. & Pogliano, K. Metabolic differentiation and intercellular nurturing underpin bacterial endospore formation. *Sci. Adv.* **7**, eabd6385 (2021).
4. Tibocho-Bonilla, J. D. *et al.* Predicting stress response and improved protein overproduction in *Bacillus subtilis*. *npj Syst. Biol. Appl.* **8**, 1–12 (2022).
5. Riley, E. P. *et al.* Spatiotemporally regulated proteolysis to dissect the role of vegetative proteins during *Bacillus subtilis* sporulation: cell-specific requirement of σ^H and σ^A . *Mol. Microbiol.* **108**, 45–62 (2018).
6. Nicolas, P. *et al.* Condition-dependent transcriptome reveals high-level regulatory architecture in *Bacillus subtilis*. *Science* **335**, 1103–1106 (2012).

REVIEWER COMMENTS

Reviewer #1 (Remarks to the Author):

The revised manuscript by Tibocho-Bonilla and colleagues has significantly improved in clarity and understandability. Many of my previous concerns have now been addressed, but others remain, especially those related to the additional experimental validation of key predictions. Additionally, access to the model and the code is still challenging, which hampers not only reproducibility but also further use of this material by the community. Altogether, I feel that at least partial resolution of these shortcomings is needed to consider the current manuscript for publication.

1. A major conclusion of this piece is that while the MC focuses on energy and amino acid production, the FS prioritizes the biosynthesis of structural components. In this context, I found the analysis of ATP production in the FS interesting; however, I really miss a detailed analysis and discussion about the production of NADPH, the other key resource for biosynthesis. How is NADPH generated in the FS? Can the authors provide a specific analysis on this topic and include the NADPH-producing pathway in Figure 3A? I consider this aspect essential for understanding the biosynthetic metabolism predicted in the FS.

We would like to thank the reviewer for pointing out the lack of discussion about a key metabolite in the energy metabolism and balance analysis. We noticed that the model predicts citrate to be fed to the FS from the MC which is then used in part of the TCA cycle to produce NADPH through CitB and Icd. This cross-feeding bypasses citrate synthase (CitZ) in the FS, which is degraded in the FS as shown already in Riley et al.¹. Notably, CitB and Icd are not essential as predicted and reported in that work. We calculated a 4% sporulation rate reduction for FS degradation of CitB and Icd if QA is inactive, and no effect if QA is active, which is in line with no sporulation defect observed in Riley et al.¹. We have added the NADPH mechanisms to Fig. 3A and discussed it in the manuscript. We would like to stress that having the available model allows the reader to evaluate any metabolite of interest.

2. As I mentioned earlier, the analysis of ATP production in the FS is very interesting, as it aligns well with previous findings and is partially validated by the high levels of Pyk and GapA observed. A side effect of this predicted glycolytic metabolism is the production of NADH. The authors subsequently suggest that LctE plays a key role as the primary regulator of NADH/NAD⁺ balancing. This is indeed an interesting hypothesis that, in my opinion, requires further exploration. In my previous report, I requested validation of some new model predictions. I understand the authors' reluctance to provide a systematic validation; however, experimentally validating the role of LctE could provide solid evidence for this predicted glycolytic metabolism in the FS by focusing on a single protein. An additional STRP and GFP-tagging analysis targeting LctE would be valuable. Also a single LctE knockout could also be interesting, as it may compromise sporulation if the authors' hypothesis is correct.

The model predicts that the MC and FS LctE are coupled, meaning that both are essential for that loop extension (pyruvate - lactate) to work. However, the model predicts LctE is not essential and that its knockout induces a reduction of 43% in the sporulation rate if QA is inactive, but it has no effect if QA is active. We performed a sporulation experiment with an LctE-null mutant, and no significant spore defect was observed (Fig. 1). Thus, the predicted mechanism is suggested as optimal but not essential for the predicted ATP supply mechanism in the FS.

Figure 1. (A) A membrane fusion assay comparing the wild type strain, PY79, to the LctE null mutant strain at 2 and 5 hours after sporulation initiation. The membrane impermeable dye, FM 4-64, in red and membrane permeable dye, MitoTracker Green, in green. No sporulation delays or engulfment defects were detected in the null mutant. Phase bright spores were present in the null mutant at levels comparable to wild type at 5 hours after sporulation initiation. (B) Spore titer comparing wild type to LctE null mutant 24 hours after sporulation initiation. Cultures were heat killed for 20 minutes at 80°C before being diluted 10-fold until 10^{-6} and plated on LB. No germination defect was detected in the null mutant.

3. When I attempted to install the software using Docker (which is an excellent choice for reproducibility), I consistently encountered an error related to the libsm1 library during the Docker build step. I tried this on multiple Linux systems with different OS configurations, but the same error occurred each time. I am wondering if this is a bug or if I need a specific installation setup or Docker version to resolve it. In my opinion, being able to reproduce the results is crucial. We thank the reviewer for identifying this issue with the Docker installation. We confirmed that libsbml is indeed the issue and is not required to run the scripts and reproduce the results, so we have taken it out of the computational environment definition. The updated repository solves this issue and all installations were successful when tested.

4. The reliance on outdated and unmaintained packages (like cobra 0.5.11) poses a significant challenge for reproducing the authors' research. I recommend updating the code or at least

providing an updated installation method for the required packages to ensure the code is reproducible.

We would like to acknowledge that the removal of cobrapy 0.5.11 makes it harder to install. However, the package is publicly available and maintained in the package's main repository (<https://github.com/opencobra/cobrapy>). The Docker contains the necessary instructions for creating the environment to reproduce our computational results.

5. Regarding the Jupyter notebooks for model generation, I noticed that both the "mother" and "spore" models have the same title. Is this an error? It's difficult to differentiate between them, as the code is nearly identical. If these models share a significant amount of code, I suggest creating a reusable function or package to consolidate the shared components, which would make the code easier to understand and maintain.

Overall, a more user-friendly installation pipeline and updated libraries are needed to ensure model reproducibility and facilitate further use

We thank the reviewer for catching this, we have corrected the error in the notebooks. Regarding the code and scripts, we agree that a user-friendly interface is needed in order for the ME-modeling work to be easier to distribute and expand to other bacteria. The revamp of ME-modeling packages and the generation of reusable code is ongoing work and will be available in an upcoming publication. Furthermore, we have indicated at the beginning of the notebooks the main differences between them to highlight spore- and mother-specific traits reconstructed. Briefly, at the stage of pre-merge (individual) model reconstruction, the main difference is the presence of the QA complex and its coupling to transport reactions in the spore model. All further modifications are described in detail in the SporeME2 reconstruction and analysis folder in analysis/spore/ (see README in the repository).

Reviewer #1 (Remarks on code availability):

3. When I attempted to install the software using Docker (which is an excellent choice for reproducibility), I consistently encountered an error related to the libsm1 library during the Docker build step. I tried this on multiple Linux systems with different OS configurations, but the same error occurred each time. I am wondering if this is a bug or if I need a specific installation setup or Docker version to resolve it. In my opinion, being able to reproduce the results is crucial.

We thank the reviewer for catching this issue, we have corrected it as explained in the previous comment.

4. The reliance on outdated and unmaintained packages (like cobra 0.5.11) poses a significant challenge for reproducing the authors' research. I recommend updating the code or at least providing an updated installation method for the required packages to ensure the code is reproducible.

We would like to acknowledge that the removal of cobrapy 0.5.11 makes it harder to install. However, the package is publicly available and maintained in the package's main repository (<https://github.com/opencobra/cobrapy>). The Docker contains the necessary instructions for creating the environment to reproduce our computational results.

5. Regarding the Jupyter notebooks for model generation, I noticed that both the "mother" and "spore" models have the same title. Is this an error? It's difficult to differentiate between them, as the code is nearly identical. If these models share a significant amount of code, I suggest creating a reusable function or package to consolidate the shared components, which would make the code easier to understand and maintain.

Overall, a more user-friendly installation pipeline and updated libraries are needed to ensure model reproducibility and facilitate further use

We thank the reviewer for catching this issue, which we addressed in the previous comment.

Reviewer #2 (Remarks to the Author):

Reviewer #2 (Remarks on code availability):

I couldn't install it with the docker instructions provided by the authors.

We thank the reviewer for identifying this issue with the Docker installation. We confirmed that libsbml is the issue and is not required to run the scripts and reproduce the results, so we have taken it out of the computational environment definition. We hope the updated repository solves this issue.

Reviewer #3 (Remarks to the Author):

I appreciate the authors' efforts to respond to my comments. All my questions and suggestions have been addressed by the authors. I have no further comments.

Reviewer #4 (Remarks to the Author):

Dear authors,

I am satisfied with the answers provided and would like to commend you on the significant improvements you have made in response to the challenges associated with the GitHub repository.

The two installation methods now offered—one traditional local installation and one using Docker—are well-considered and provide a valuable balance between maintaining consistency with established protocols and ensuring ease of use. The traditional approach, while requiring several manual steps, is aligned with practices seen in other ME-model projects, and I appreciate that you have kept it for users who prefer a more hands-on approach. Your transparency regarding the necessity to manually install COBRAPy 0.5.11, despite its removal from PyPI, is especially commendable, as it ensures users are fully aware of potential challenges in advance.

The Docker-based solution is a major enhancement. By offering also this more streamlined option, you have significantly reduced the complexity of setting up the computational environment, making it much more accessible to a wider audience. The clear and well-documented instructions for creating a Docker image and running a container, along with the Jupyter Notebook session that simplifies result reproduction, demonstrate your commitment to making the research more reproducible and user-friendly.

Overall, your efforts to improve usability, enhance reproducibility, and provide clear documentation have substantially strengthened the quality and impact of this study. I appreciate the time and care you've invested in addressing reviewer feedback, and these changes undoubtedly make the work more robust and accessible to users across diverse areas of bacterial spore studies and beyond.

Reviewer #4 (Remarks on code availability):

I indeed (now) managed to get the code running.

Reviewer #5 (Remarks to the Author):

I would like to thank the authors for their careful consideration of the comments. I have reviewed the revised manuscript and the code and I am pleased to see that the authors have extensively addressed the feedback in a thorough and thoughtful manner. The revisions have strengthened the manuscript, and the clarifications/corrections provided improve both the quality of the work and the reproducibility of their results.

I have just one follow-up comment regarding the authors' response to comment 2. Once this point has been addressed, and given the positive incorporation of the rest of the feedback, I will consider the manuscript acceptable for publication.

Regarding comment 2 from the first revision:

2. Following the previous comment, the authors seem to tackle that matter by only considering growth of the forespore in SporeME2 model. The code shows a spore biomass reaction and a section that mentions 'Mother is not growing' (BuildCommunityModel.ipynb). Considering the mother cell produces the main building blocks (biomass precursors) for the forespore, which is a sign of cell growth, it is not clear how the authors represent growth (if represented) on the mother cell. Does the mother cell have a separate biomass reaction? How is it defined and constrained in the model? The authors need to clarify this point in the text and justify their modeling framework.

We thank the reviewer for catching the lack of description of the objective function and the solution strategy in the manuscript. Precisely, both cells present growth rates, as the reviewer correctly inferred, because they produce expression machinery, which must be diluted to biomass in order to keep the steady state. However, the key simplification that allowed us to solve this model is that only the forespore's growth rate is optimized, giving an optimization problem that is still non-linear but only one-dimensional. In this way, we were able to use the bisection method described previously in the solver package, solveME. We have now included a paragraph in the methods providing these details, and have also updated the notebook's misleading details. The paragraph reads:

"Regarding the second question, our model only optimizes for the formation rate of the forespore, which is assumed to be the only objective function. Therefore, SporeME2 was defined based on only the growth rate of the forespore (forespore formation rate). This simplification was critical in solving SporeME2, as it reduces the two-dimensional non-linear programming problem (with two variable growth rates) to a one-dimensional non-linear programming problem, which can be solved using the bisection method described in the ME-model solver package, solveME."

The authors have clarified that, to simplify the modeling approach, they only account for the growth rate of the foreshore as the community objective function. However, there is also a biomass reaction representing the growth of the mother cell. Considering that this is not part of the objective function, the authors need to explicitly clarify in the text how they are constraining the growth rate of the mother cell (presumably to 0). This clarification is important to ensure the model's assumptions and structure are fully transparent.

We thank the reviewer for identifying this lack of explanation on how the growth rate of the mother cell is treated in our simulations. The reviewer is correct in that, to carry any metabolic flux, the mother cell needs to synthesize enzymatic machinery, which necessarily contributes to biomass according to the ME-model formulation. Because of this, we allowed for biomass production flux in the mother cell to allow for feasibility and to close the mass balance in the steady state, but we only optimized for the forespore biomass production flux. We have clarified this in the manuscript's methods.

Reviewer #5 (Remarks on code availability):

The installation steps and the content of the repository is much more clear. I appreciate the authors effort to include a docker container. I have installed the docker container and I have run several Jupyter notebooks and scripts. Everything I tried seemed to run nicely.

References

1. Riley, E. P., Lopez-Garrido, J., Sugie, J., Liu, R. B. & Pogliano, K. Metabolic differentiation and intercellular nurturing underpin bacterial endospore formation. *Sci. Adv.* **7**, eabd6385 (2021).